# Coupling of Polo kinase activation to nuclear localization by a bifunctional NLS is required during mitotic entry

David Kachaner[1,2], Damien Garrido[1,2], Haytham Mehsen[1], Karine Normandin[1], Hugo Lavoie[1] & Vincent Archambault[1,2]

The Polo kinase is a master regulator of mitosis and cytokinesis conserved from yeasts to humans. Polo is composed of an N-term kinase domain (KD) and a C-term polo-box domain (PBD), which regulates its subcellular localizations. The PBD and KD can interact and inhibit each other, and this reciprocal inhibition is relieved when Polo is phosphorylated at its activation loop. How Polo activation and localization are coupled during mitotic entry is unknown. Here we report that PBD binding to the KD masks a nuclear localization signal (NLS). Activating phosphorylation of the KD leads to exposure of the NLS and entry of Polo into the nucleus before nuclear envelope breakdown. Failures of this mechanism result in misregulation of the Cdk1-activating Cdc25 phosphatase and lead to mitotic and developmental defects in *Drosophila*. These results uncover spatiotemporal mechanisms linking master regulatory enzymes during mitotic entry.

[1] Institute for Research in Immunology and Cancer, Université de Montréal, C.P. 6128 Succursale Centre-Ville, Montréal, QC, Canada H3C 3J7. [2] Département de biochimie et médecine moléculaire, Université de Montréal, C.P. 6128 Succursale Centre-Ville, Montréal, QC, Canada H3C 3J7. Correspondence and requests for materials should be addressed to V.A. (email: vincent.archambault.1@umontreal.ca)

Cell division requires the spatiotemporal coordination of many intracellular events that are coordinated by various master regulator enzymes[1]. Some of these enzymes control multiple events and substrates, and to this end, they are themselves subject to spatiotemporal regulation. As a result, an intricate regulatory network of enzymes has evolved to control the cell division cycle. Mechanistically, how the coordination of localization and activity is achieved for each enzyme during the cell cycle is incompletely understood.

The spatiotemporal regulation of Polo kinase (Polo-like kinase 1/Plk1 in humans) is a particularly important and difficult case to solve. Discovered in *Drosophila*, Polo is the founding member of the polo-like kinase (PLK) family[2–4]. Polo homologs in eukaryotes control myriad events in cell division, from mitotic entry to cytokinesis[5, 6]. Polo phosphorylates serine and threonine residues on numerous substrates to modify their activities, at several locations in the cell.

The localization of Polo changes during the cell cycle[7–9]. In interphase, Polo shows a mostly diffuse cytoplasmic localization. Polo becomes concentrated on centrosomes from early prophase and appears on centromeres or kinetochores from late prophase, before nuclear envelope breakdown (NEB). After anaphase onset, Polo relocalizes to the central spindle and remains enriched at the midbody ring in late stages of cytokinesis. Polo localization at these structures requires the polo-box domain (PBD), a C-terminal protein interaction domain that defines the PLK family[10, 11]. Docking of the PBD to cellular structures or to individual substrates facilitates their phosphorylation by the kinase domain (KD) of Polo[12]. The affinity of many mitotic targets for the PBD is strongly enhanced by priming phosphorylation events, often at Cdk1 or Polo consensus sites[11].

The activities of the KD and PBD of Polo are regulated during cell division by an intramolecular mechanism governed by post-translational modifications and protein interactions[13]. The PBD and the KD interact with and inhibit each other[14–16]. Although disfavored when Polo is inactive, binding of the PBD to a target contributes to relieve KD inhibition[14]. Conversely, phosphorylation of the KD at the activation loop was proposed to promote the dissociation of the KD from the PBD, although the precise mechanism is unclear[15, 17, 18].

Work in *Drosophila* has facilitated the study of mechanisms of Polo regulation during mitosis and cytokinesis. In this system, the inactive conformation of Polo, in which the PBD and KD interact, is stabilized by Map205, a PBD-interacting protein that sequesters Polo on microtubules during interphase[17, 19]. Phosphorylation of the Polo kinase at its activation loop by Aurora B promotes the release of Map205 from the PBD, favoring binding of the PBD to its targets on centrosomes, kinetochores, and the midbody during cytokinesis[18]. Although no vertebrate ortholog of Map205 is known, the general mechanism of interdomain regulation of Polo appears to be largely conserved[13].

How Polo is regulated for its earliest functions, during mitotic entry, remains incompletely understood. In all systems examined, Polo and its orthologs promote mitotic entry by facilitating the activation of Cyclin B-Cdk1, which phosphorylates multiple substrates to promote chromosome condensation, NEB, and spindle formation[8]. Cdk1 is initially kept inactive by phosphorylation at Thr14 and Tyr15 by the Wee1 and Myt1 kinases, and Cdk1 becomes activated by Cdc25 phosphatases that dephosphorylate these sites[1]. In vertebrates, phosphorylation of Wee1 and Myt1 by Plk1 contributes to their inactivation[20, 21]. Conversely, phosphorylation of Cdc25C by Plk1 contributes to its activation[22]. In vertebrates, Plk1 kinase activity increases in G2 by phosphorylation of its activation loop. This event is mediated by the Aurora A kinase and its co-factor Bora[23, 24].

Plk1 is targeted to centromeres as early as G2 by an interaction with PBIP1[25]. In *Drosophila*, Polo also appears on centromeres before NEB, where it interacts with INCENP, a component of the Chromosomal Passenger Complex[26]. During prophase, Polo/Plk1 also localizes to centrosomes and plays essential roles in centrosome maturation[7, 27]. Thus, Polo/Plk1 localizes to both the cytoplasm and the nucleus in prophase. A nuclear localization signal (NLS) has been identified in Plk1 but how it regulates Plk1 in prophase is unknown[28]. Intriguingly, while the phospho-activated form of Plk1 first appears on centrosomes in G2, Plk1 activity is first detected in the nucleus during mitotic entry, and Plk1 nuclear localization is required for the G2/M transition in recovery from the DNA damage response[23, 29]. How Polo/Plk1 activation and nucleocytoplasmic localization are coordinated during mitotic entry is unknown.

Here, using the *Drosophila* model system, we have elucidated how Polo activation is coupled to its nuclear localization in prophase. We found that the NLS motif of Polo mediates the interdomain inhibitory interaction in interphase. Activating phosphorylation of the KD during mitotic entry induces the exposure of the NLS and relocalization of Polo to the nucleus. We explored the functional consequences of disrupting this mechanism at the molecular, cellular, and organism levels.

## Results

**Phosphorylation of Polo triggers its entry into the nucleus.** In order to evaluate the contributions of different regulation mechanisms on Polo activity during the cell cycle in vivo, we constructed transgenic flies allowing inducible expression of various versions of Polo fused C-terminally to GFP (*UASp-POLO-GFP*). The Polo variants were either wild-type (WT) or contained amino-acid substitutions disrupting its activities, protein interactions, or phosphorylation (Supplementary Fig. 1a). We expressed these proteins in early embryos, where nuclei divide rapidly in a syncytium. Live imaging revealed that Polo^WT-GFP is enriched at centrosomes, nuclei, and centromere/kinetochore regions in mitosis, and at the central spindle and midbody ring in karyokinesis (Fig. 1a; Supplementary Movie 1). This localization pattern is seemingly identical to that previously reported for Polo fused to GFP at the N-terminus, suggesting that it reflects the normal localization of Polo[19, 30]. To test the importance of PBD function for this localization pattern, we mutated residues analogous to those required for phospho-binding in human Plk1 (W395F, H518A, K520A = Polo^PBDmut)[14]. Polo^PBDmut-GFP fails to localize to kinetochores and the midbody, consistent with previous work showing that Plk1 and orthologs require prior phosphorylation of their targets to localize at these sites[8] (Supplementary Fig. 1b). The localization of Polo^PBDmut-GFP to centrosomes is reduced (relative to the nuclear signal) but not completely eliminated, consistent with previous findings with human Plk1[31]. However, Polo^PBDmut-GFP localizes to the nucleus normally, suggesting that Polo does not require binding to a phosphorylated target to enter the nucleus.

As previously found in cell culture, replacing the activation loop phosphorylation site in the KD, Thr182, with an alanine residue (T182A) prevents the recruitment of Polo-GFP to the midbody in embryos[18] (Fig. 1a; Supplementary Movie 2). Unexpectedly, we found that the T182A mutation strongly decreases the nuclear localization of Polo-GFP from karyokinesis to prophase (Fig. 1a; Supplementary Fig. 1c). This effect cannot be attributed to a lower kinase activity because a kinase-dead mutant (K54M, Polo^kin.dead-GFP) localizes normally to nuclei (Supplementary Fig. 1d). The localization of a phosphomimetic mutant Polo^T182D-GFP is similar to Polo^WT-GFP, including in nuclei (Fig. 1a; Supplementary Fig. 1c).

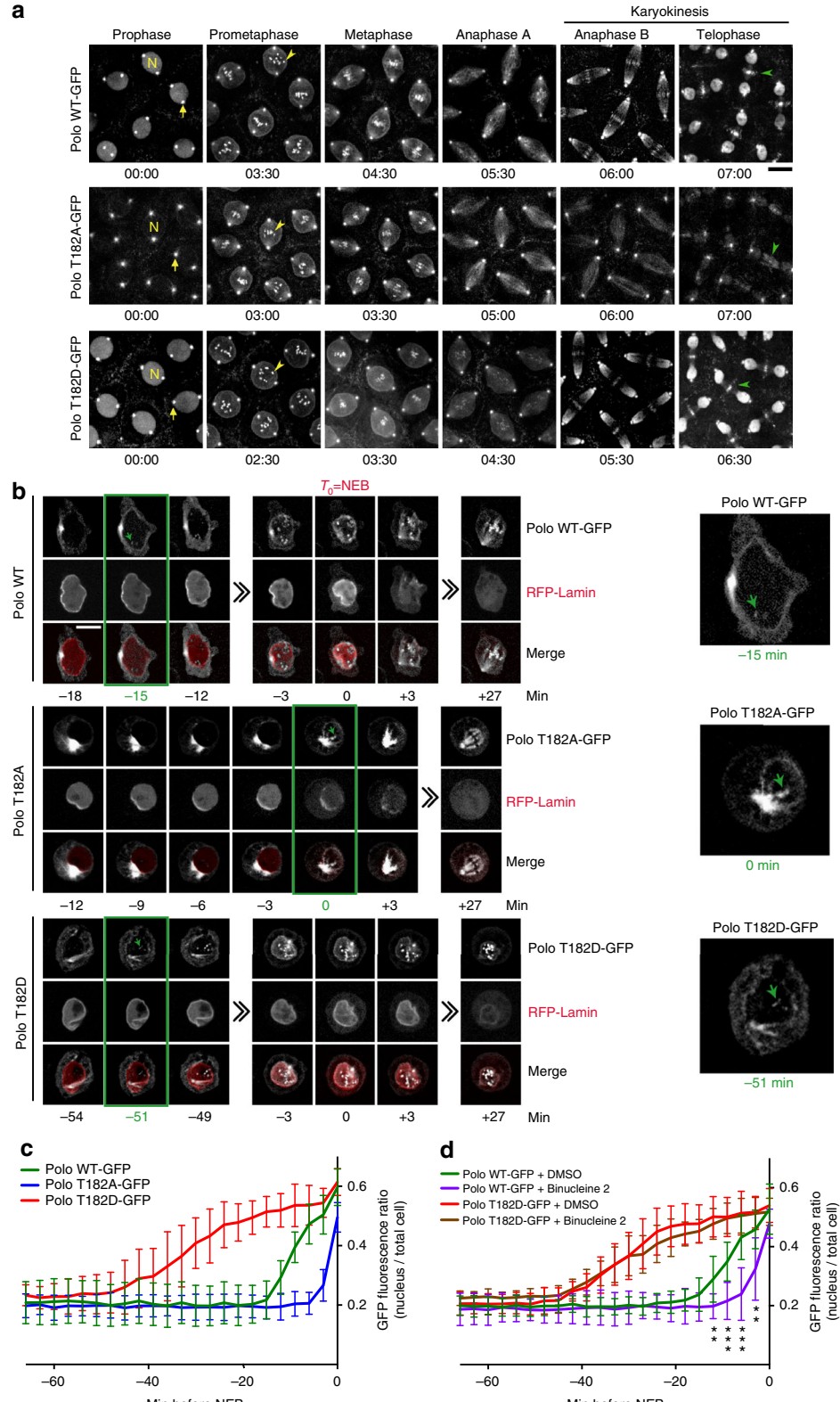

**Fig. 1** The activating phosphorylation site in the kinase domain of Polo controls its nuclear localization. **a** Syncytial embryos expressing Polo^WT-GFP, Polo^T182A-GFP, or Polo^T182D-GFP were observed by time-lapse microscopy. N: a nucleus; arrow: a centrosome; yellow arrowhead: centromere/kinetochore; green arrowhead: midbody ring. Bar: 10 μm. **b** Time-lapse imaging of stable cell lines expressing Polo^WT-GFP, Polo^T182A-GFP, or Polo^T182D-GFP with RFP-Lamin. The time between the initial detection of Polo-GFP at centromeres (green arrows in green box and enlarged images, right) and NEB ($T_0$) is shown. NEB was determined when RFP-Lamin starts to dissolve into the cytoplasm. Bar: 5 μm. **c** Quantification of the GFP fluorescence ratio (nucleus/total cell) in the period preceding NEB, from experiments in **b**. ($n = 10$, error bars: SD). **d** Quantification following treatment with Binucleine 2, as in **c**. See Supplementary Fig. 2a–d for images. **P < 0.01; ***P < 0.001; Student's $t$ test, between Polo^WT-GFP + DMSO vs. Binucleine 2

The observed effect of the T182A mutation on Polo localization to the nucleus could have been specific to embryonic cell cycles, which are extremely rapid and synchronized without Gap phases between S and M phases. To examine if it also occurred in a normal cell cycle, we generated stable D-Mel2 cell lines allowing copper-inducible expression of the different forms of Polo-GFP. In these cells, Polo$^{WT}$-GFP is mostly cytoplasmic in

interphase, and it begins to appear in the nucleus during prophase, localizing to centromeres/kinetochores around 15 min before NEB as detected by the dissolution of RFP-Lamin (Fig. 1b, c; Supplementary Movie 3). As in embryos, the T182A mutation prevents the nuclear localization of Polo-GFP, the Polo$^{T182A}$-GFP protein appearing at kinetochores only after NEB (Supplementary Movie 4). We have previously shown that in *Drosophila*,

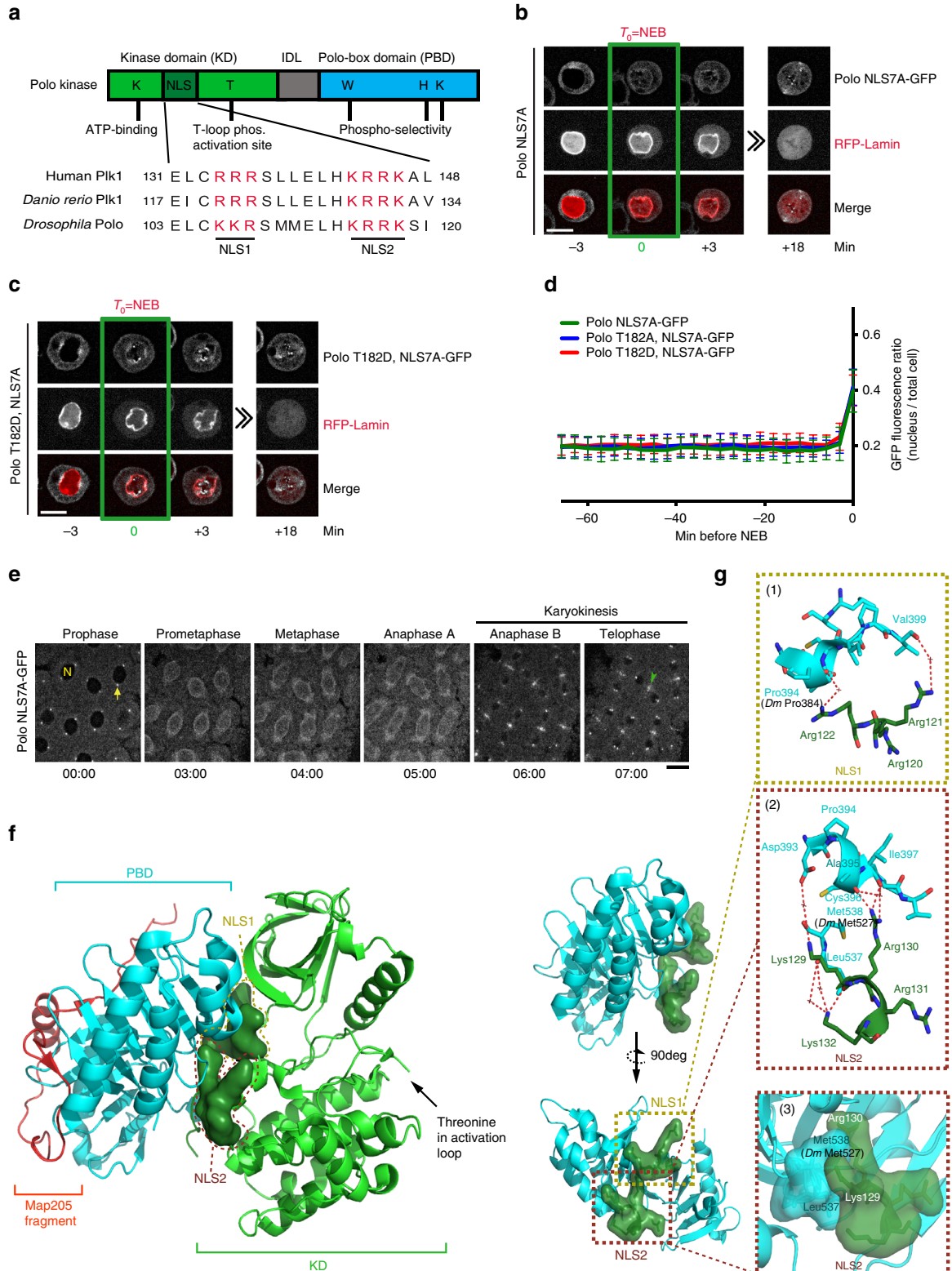

phosphorylation of Polo at T182 is primarily dependent on Aurora B kinase[26]. Consistent with this, treatment with Binucleine 2, a selective inhibitor of *Drosophila* Aurora B, delays Polo^WT^-GFP entry into the nucleus in prophase (Fig. 1d; Supplementary Fig. 2a, b, e). On the other hand, the phosphomimetic Polo^T182D^-GFP mutant accumulates in the nucleus ~25 min earlier than Polo^WT^-GFP, even in the presence of Binucleine 2 (Fig. 1b–d; Supplementary Fig. 2c, d; Supplementary Movie 5). This effect was not due to an increase in Polo kinase activity because the nuclear accumulation of a Polo^T182D, kin.dead^-GFP is also advanced relative to Polo^kin.dead^-GFP (Supplementary Fig. 3). Altogether, these results suggest that activation loop phosphorylation of Polo triggers its entry into the nucleus.

**A NLS is required for Polo to enter the nucleus in prophase.** About 15 years ago, an NLS was reported in human Plk1 and it is conserved in *Drosophila*[28] (Fig. 2a). We found that mutation of seven NLS residues into alanine residues (NLS7A) abolished the nuclear localization of Polo in prophase in both cultured cells and embryos (Fig. 2b–e; Supplementary Fig. 1c; Supplementary Movie 6). Moreover, adding the T182D mutation in Polo^NLS7A^-GFP did not promote its nuclear/centromeric localization during prophase and adding T182A had no effect either (Fig. 2c, d; Supplementary Fig. 4a, b). In embryos, Polo^NLS7A^-GFP localized more strongly around the mitotic spindle—possibly on membranous structures—rather than on kinetochores and centrosomes, for unknown reasons (Fig. 2e).

To investigate how activation loop phosphorylation and the NLS might function together in regulating the nuclear localization of Polo, we examined the recently published crystal structure of a complex between the PBD and KD of zebrafish Plk1, stabilized by a peptide from *Drosophila* Map205[17]. This analysis revealed that the NLS is composed of residues in the hinge connecting the N- and C-lobes of the KD, a region that directly contacts the PBD (Fig. 2f, g). Thus, binding of the PBD to the KD apparently masks the NLS. Consistent with this idea, exposing the NLS motif of Polo by fusing it to the N-terminus of Polo^WT^-GFP or Polo^NLS7A^-GFP made both proteins constitutively nuclear, as verified by microscopy and cellular fractionation (Supplementary Fig. 4c, d). The interaction between the PBD and the KD is known to mediate a reciprocal inhibition, whereby the PBD blocks KD catalysis and the KD keeps the PBD in a conformation that has a lower affinity for phospho-primed targets[17, 18]. Moreover, it has been proposed that phosphorylation of the KD at the activation loop could induce the dissociation of both domains[15, 18]. Together, these observations suggest a model in which phosphorylation-induced interdomain dissociation exposes the NLS and triggers nuclear import of Polo, thus effectively coupling Polo activation and nuclear localization during mitotic entry.

**The phospho-site and the NLS control the KD–PBD interaction.** We tested the above model biochemically. We produced a GST–PBD fusion in bacteria and used them in pulldown experiments with extracts from cells expressing different versions of Flag-tagged Polo KD. As expected, Flag-KD could interact with GST–PBD and the T182A mutation had no obvious effect, while the T182D strongly diminished the interaction (Fig. 3a). In view of the published crystal structure, we reasoned that the NLS7A mutation was likely to abrogate the KD–PBD interaction (Fig. 2f, g). As predicted, we found that this mutation completely disrupts the KD–PBD complex in the GST pulldown assay (Fig. 3a).

We previously showed that Polo interacts with Map205 on microtubules in interphase[19]. Unlike most PBD targets, Map205 does not require prior phosphorylation as it comprises a naturally phosphomimetic site. In addition, while most targets are thought to bind the PBD in its active conformation, where the PBD is free from inhibition by the KD, Map205 binds the PBD preferentially in its inactive conformation, where it is bound and inhibited by the KD[13, 17, 18]. We made used the Polo-Map205 interaction to further test our model. As previously found, the T182D mutation in Polo weakens its interaction with Map205, while the T182A mutation has no detectable effect in co-purification assays (Fig. 3b). In agreement with our model, the NLS7A mutation in the KD, which disrupts the KD–PBD interaction, abrogates the PBD-dependent interaction of Polo with Map205. These biochemical results are consistent with the localization of the different forms of Polo in cells (Supplementary Fig. 5). Map205 mediates localization of a pool of Polo-GFP to microtubules and away from the midbody ring in cytokinesis[18]. As shown previously, the T182A mutation enhances the Map205-dependent sequestration of Polo-GFP to microtubules and prevents its localization to the midbody ring in cytokinesis[18]. Conversely, the NLS7A mutation disrupts the localization of Polo-GFP to microtubules and promotes its targeting to the midbody ring, even when combined with the T182A mutation (Supplementary Fig. 5). Together with the findings described above, these results suggest that phosphorylation of Thr182 in the KD leads to a structural change in Polo that disrupts its NLS-dependent interaction of the KD with the PBD and promotes dissociation of the PBD from Map205.

To test the regulation of the KD–PBD interaction in living cells, we used bioluminescence resonance energy transfer (BRET)[32]. In this assay, the PBD of Polo was expressed in fusion with GFP and the KD was expressed in fusion with Luciferase (Luc). When the two proteins interact, energy is transferred from Luc to GFP, which then emits fluorescence measured as BRET (Fig. 3c). Cells were transfected with fixed amounts of vectors for expression of different forms of Luc-KD (donor) and with increasing amounts of vectors expressing PBD–GFP (acceptor). A Map205 peptide which promotes the KD–PBD interaction was also expressed. With KD^WT^, the BRET steeply increases toward a plateau as a function of acceptor/donor ratio, indicating a specific interaction. By comparison, BRET signals with KD^NLS7A^ increases almost linearly, indicating abrogation of specific KD–PBD interaction. While the curve obtained with the KD^T182A^ is not significantly different from

**Fig. 2** Nuclear accumulation of Polo depends on an NLS that is masked by the PBD. **a** Alignment of Polo sequences from different species reveals a conserved bipartite NLS in the KD comprising the NLS1 and NLS2 motifs. **b, c** Time-lapse imaging of stable cell lines expressing Polo^NLS7A^-GFP and Polo^T182D, NLS7A^-GFP with RFP-Lamin. Bar: 5 μm. **d** The GFP fluorescence ratio (nucleus/total cell) was measured in the period preceding NEB (*n* = 10, error bars: SD). **e** A syncytial embryo expressing Polo^NLS7A^-GFP was observed by time-lapse microscopy. N: a nucleus; arrow: a centrosome; arrowhead: midbody ring. Bar: 10 μm. **f** The NLS in the crystal structure of a complex between the KD (green) and the PBD (cyan) of zebrafish Plk1 and the inhibitory peptide from *Drosophila* Map205 (red) (Protein Data Bank accession no. 4J7B[17]). The surface area surrounding NLS1 and NLS2 residues is shown (dark green) to highlight their interaction with the PBD. Close-ups show the interface between NLS residues and the PBD (top) and the same surface with a clockwise 90° rotation (bottom). **g** Dashed boxes emphasize details of the interaction between NLS1 and a PBD peptide in zebrafish Plk1 (1) as well as hydrogen bonds (2) and hydrophobic packing (3) between NLS2 and the PBD. Residues participating at the interface are drawn in sticks and are labeled. Equivalent residues in *Drosophila* Polo that were mutated in this study (Fig. 3) are in parentheses. Hydrogen bonds are indicated by red dashed lines. Intervening water molecules are indicated by a cross. Structural rendering was performed using PyMOL 1.4 built-in commands

the one obtained with KD$^{WT}$, the KD$^{T182D}$ results in an intermediate curve between the KD$^{WT}$ and KD$^{NLS7A}$. This result further supports the conclusion that phosphorylation of Polo at T182 leads to a decrease in affinity between the KD and the PBD and that the NLS is required for KD–PBD complex formation.

**The NLS-dependent interdomain interaction inhibits Polo.** Since binding of the PBD to the KD inhibits its activity and since NLS residues are required for the PBD to interact with the KD, we hypothesized that mutation of the NLS might increase kinase activity. In in vitro kinase assays, we found that the NLS7A

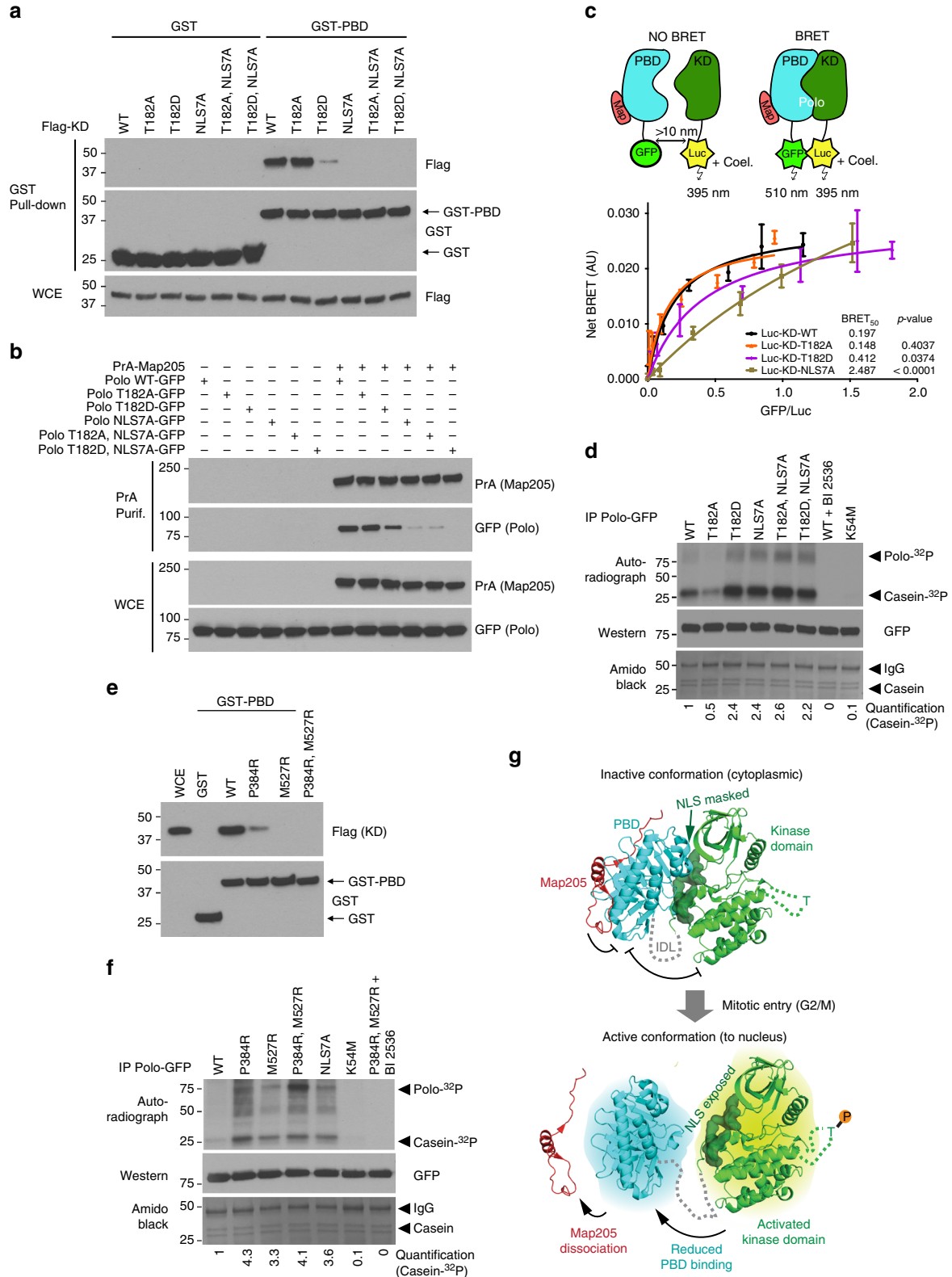

mutation more than doubles the activity of immunoprecipitated Polo-GFP (Fig. 3d). The activity of Polo$^{NLS7A}$-GFP is ~5-fold higher than non-phosphorylatable Polo$^{T182A}$-GFP, and very similar to the activity of Polo$^{T182D}$-GFP. Combining the T182D and NLS7A mutations does not further increase Polo kinase activity relative to single mutants. Moreover, Polo$^{T182A, NLS7A}$-GFP has a similar activity to Polo$^{NLS7A}$-GFP. Finally, mutations of PBD residues (P384R, M527R) that interact directly with NLS residues of the KD (Fig. 2f, g) abrogate the PBD–KD interaction and increase Polo kinase activity to a similar level as that of Polo$^{NLS7A}$ or Polo$^{T182D}$ (Fig. 3e, f). These results indicate that the KD of Polo can be fully activated when relieved from PBD-dependent inhibition, even when it is not phosphorylated in its activation loop. It also suggests that the main function of activation loop phosphorylation is to relieve inhibition of the KD by the PBD. As a result of this activation, the NLS of Polo is concomitantly exposed (Fig. 3g). This molecular mechanism effectively couples Polo activation to its nuclear localization in prophase.

**Nuclear localization of Polo positively regulates Cdc25.** We sought to determine if the nuclear targeting of Polo in prophase by the above mechanism is required for its function during mitotic entry. In vertebrates, Cdc25 phosphatases are crucial targets of Plk1 and their phosphorylation contributes to their activation, enabling the removal of inhibitory phosphorylation sites on Cdk1, which triggers mitotic entry (Supplementary Fig. 6a)[33]. *Drosophila* somatic cells express a single Cdc25 named String[34]. We examined the localization of GFP-Cdc25$^{String}$ using stable cell lines also expressing RFP-Lamin. We found that GFP-Cdc25 is nuclear during interphase and is relocalized to the cytoplasm during prophase, becoming excluded from the nucleus between 15 and 10 min before NEB (Fig. 4a–c; Supplementary Movie 7). Interestingly, this event coincides with the relocalization of Polo to the nucleus before NEB (Fig. 4b, c; Supplementary Movie 8). To test if Polo activity is required for the nuclear exclusion of Cdc25, we inhibited Polo with BI 2536[18, 35]. As a result, GFP-Cdc25 is no longer excluded from the nucleus; instead it gradually becomes dispersed throughout the cell over more than 1 h (Supplementary Fig. 6b, c). This result suggests that Polo activity is required for the rapid relocalization of Cdc25 from the nucleus to the cytoplasm in prophase. Interestingly, we found that inhibition of Polo with BI 2536, in addition to causing increases in mitotic index and abnormal mitotic spindles, hampers the disassembly of the nuclear envelope visualized by Lamin staining (Supplementary Fig. 6d–g). As Lamin disassembly depends on Cdk1 activity in various systems[36], defects in this process could be due to a failure of Cdc25 to fully activate Cdk1. To test if the targeting of Polo to the nucleus by the mechanism we uncovered is sufficient to induce the relocalization of Cdc25,

we transfected cells with Myc-Cdc25 alone or with the different variants of Polo-GFP, and then we subjected cells to subcellular fractionation and immunofluorescence (IF). In both assays, Myc-Cdc25 is a predominantly nuclear protein in interphase (Fig. 4d; Supplementary Fig. 6h). Expression of Polo$^{WT}$-GFP or Polo$^{T182D}$-GFP induces the relocalization of Myc-Cdc25 to the cytoplasm, accompanied by an upshift in the apparent molecular mass of Cdc25, likely due to increased phosphorylation (Fig. 4d; Supplementary Fig. 6i). Cdc25C is known to migrate more slowly on a gel when phosphorylated by Plk1[22]. Both the relocalization and mobility shift of Cdc25 required Polo kinase activity as they were abolished by treatment with BI 2536. By contrast, expression of Polo$^{T182A}$-GFP or Polo$^{NLS7A}$-GFP does not change the localization or mobility of Cdc25, despite the latter mutant having increased kinase activity compared to Polo$^{WT}$-GFP. We conclude that Polo activity in the nucleus is necessary and sufficient for Cdc25 relocalization from the nucleus to the cytoplasm.

To assess if the Polo-dependent spatial regulation of Cdc25 impacts on Cdk1 activation, we probed phosphorylation levels of Cdk1 at Tyr15, an inhibitory site dephosphorylated by Cdc25. Overexpression of Polo$^{WT}$-GFP or Polo$^{T182D}$-GFP results in lower phosphorylation levels of Cdk1 at Tyr15 (Fig. 4e). In contrast, Polo$^{T182A}$-GFP and Polo$^{NLS7A}$-GFP have no effect on pTyr15. These results suggest that Polo phosphorylates Cdc25 in the nucleus to induce the relocalization of Cdc25 to the cytoplasm, where it dephosphorylates and activates Cdk1.

We then tested the effect of Polo activity on Cdc25 phosphatase activity. Different forms of Polo-GFP were transfected into cells and co-expressed Myc-Cdc25 (WT or phosphatase-dead) was immunoprecipitated before its phosphatase activity was tested in vitro on a Cdk1 peptide phosphorylated at Tyr15. We found that Cdc25 activity is higher after transfection of active forms of Polo that can localize to the nucleus (WT and T182D) compared with forms of Polo that do not localize to the nucleus (Fig. 4f). Altogether, these results suggest that Polo phosphorylation of Cdc25 triggers its nuclear export and activation.

Consistent with direct regulation of Cdc25 by Polo in the nucleus, Myc-Cdc25 co-immunoprecipitates strongly with Polo$^{WT}$-GFP, but more weakly with Polo$^{NLS7A}$-GFP (Fig. 4g). Interestingly, we also found that Cdk1 activity contributes to reinforce this regulation as inhibition of Cdk1 with RO 3306 abrogates the Polo-Cdc25 interaction and the mobility shift of Cdc25. These observations are consistent with previous demonstrations that phosphorylation of Cdc25B and Cdc25C by Cdk1 enhances their binding and phosphorylation by Plk1 in human cells[11, 37].

**Coordination of Polo activity is required for mitosis.** To examine the importance of the spatiotemporal coordination of Polo for cell division, we generated stable cell lines allowing the

**Fig. 3** Interdependence between Polo kinase activity, NLS function, and PBD function. **a** Mutation of NLS residues or T182D substitution abrogates the KD–PBD interaction. GST–PBD or GST-bound sepharose beads were incubated with lysates of cells transfected with different forms of Flag-KD (with the IDL) as indicated. Pulldown products were analyzed by western blots. **b** Mutation of NLS residues or T182D substitution abrogates the Polo-Map205 interaction. Cells were transfected as indicated and PrA-Map205 was purified. Samples were analyzed by western blotting. **c** Bioluminescence resonance energy transfer (BRET) reveals the impact of T182D and NLS7A mutations on the KD–PBD interaction in live cells. HEK293T cells were transfected with a fixed amount of Luc-KD (with the IDL) expression vector and increasing amounts of PBD–GFP expression vector. A third plasmid expressing a Map205 fragment (Map), which stabilizes the PBD–KD complex was co-transfected. When the KD and PBD interact, the luciferase (Luc) moiety, upon reaction with coelenterazine, transfers energy to GFP, which then fluorescences (BRET). Differences in BRET$_{50}$ (the GFP/Luc ratio at which BRET is half-maximal) reflect differences in affinity. AU arbitrary units. Error bars: standard deviation of triplicate values from a representative experiment. **d** Mutation of NLS residues increases Polo kinase activity. Immunoprecipitated Polo-GFP (WT and mutants) were used in kinase reactions using casein as a substrate. For Polo inhibition, BI 2536 was added at 300 nM. Reactions were analyzed by autoradiography, western blots, and amido black (total protein). **e** Mutation of NLS-interacting PBD residues prevents the KD–PBD interaction. Experiment as in **a**. **f** Mutation of NLS-interacting PBD residues increases Polo kinase activity. Experiment as in **d**. **g** Model for coupling of Polo activation and nuclear localization. See text for details. The crystal structure of a complex between the KD (green) and the PBD (cyan) of zebrafish Polo and the inhibitory peptide from *Drosophila* Map205 (red) (Protein Data Bank accession no. 4J7B[17]) was used for structure rendering with PyMOL 1.4

inducible expression for different forms of Polo-GFP. Endogenous Polo could be simultaneously silenced by RNAi targeting its 3′ untranslated region (UTR) (Fig. 5a). In these cells, H2A-RFP was expressed as a chromosome marker. We used live imaging to follow cell division. RNAi silencing of Polo results in mitotic

failures[38]. We previously observed that expression of Polo^WT-GFP complements the loss of endogenous Polo[26] (Fig. 5b–e). In contrast, expression of Polo^NLS7A-GFP fails to complement the depletion of endogenous Polo. The time between late prophase (approximately determined by a sudden increase in chromosome

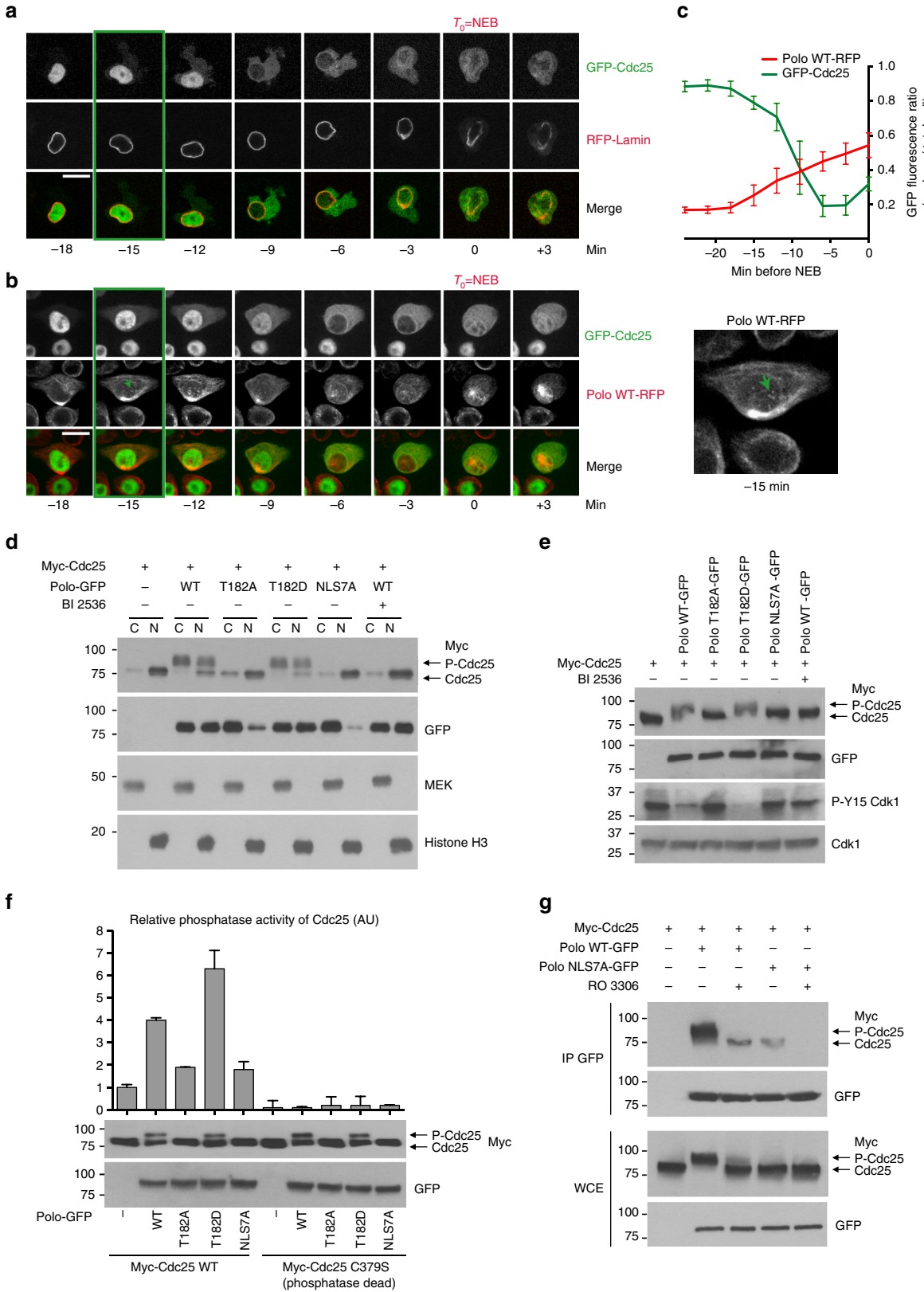

condensation) and anaphase doubles when Polo[NLS7A]-GFP is expressed, compared with Polo[WT]-GFP (Fig. 5b, c). This effect is accompanied by an increase in lagging chromosomes and cytokinesis failures (Fig. 5d, e). The mitotic delay and cytokinesis defects seen upon expression of Polo[NLS7A]-GFP are recessive relative to endogenous Polo. This suggests that they result from the loss of Polo function in the nucleus rather the gain of Polo kinase activity in the cytoplasm caused by the NLS7A mutation. Consistent with this idea, expression of the constitutively active Polo[T182D]-GFP, which is competent for nuclear translocation, does not cause significant mitotic delays or cytokinesis defects. On the other hand, an increase in lagging chromosomes occurs when Polo[NLS7A]-GFP is expressed, even in the presence of endogenous Polo, suggesting that this defect may be partly attributed to an increase in Polo kinase activity. Finally, we tried to image mitosis in cells expressing constitutively nuclear NLS-Polo[WT]-GFP or NLS-Polo[NLS7A]-GFP (Supplementary Fig. 4c) but these cells never entered mitosis. Although we do not understand the underlying mechanism, this observation suggests that constitutively high nuclear levels of Polo interfere with the cell cycle, imposing a need for restricted nuclear import of Polo in interphase.

**Coordination of Polo activity is required in vivo.** We used *Drosophila* genetics to examine the importance of the coordination between Polo activity and localization in vivo. We found that silencing Polo expression in developing head tissues, using the *eyeless-Gal4* driver and a *UAS-Polo* RNAi construction, results in headless flies and failure to hatch after pupation, with complete penetrance (Fig. 6a; Supplementary Fig. 7a). We assayed the ability of our different Polo-GFP transgenes to rescue head development in this system (Supplementary Fig. 7b, c). Expression of the different transgenes was driven simultaneously to the RNAi construction using *eyeless-Gal4*. The different forms of Polo-GFP were made insensitive to RNAi by synonymous codon replacements in the complimentary DNA. In this assay, Polo[WT]-GFP largely rescues head development, although heads and eyes are often smaller for unknown reasons (Fig. 6a, b; Supplementary Fig. 7d–f). By contrast, the kinase-dead and PBD mutants fail to rescue head development, indicating that KD activity and target binding by the PBD are both required for Polo function in vivo. Polo[T182A]-GFP similarly fails to rescue head development, indicating that Polo activation by phosphorylation at the activation loop or nuclear translocation of Polo in prophase or both is essential in vivo. For all mutants, a small fraction of flies hatch with apparently normal heads (Fig. 6b). We do not know how this infrequent escaping happens.

To assess if the NLS motif is required for Polo function in vivo, we tested the ability of Polo[NLS7A]-GFP to rescue head development. We found that it fails to rescue viability as scored by pupae hatching (Fig. 6b). However, we observed a partial rescue in head

development in 17% of the pharate adults examined (Fig. 6a). Thus, NLS residues are essential for normal Polo function in vivo. Although Polo[NLS7A]-GFP fails to localize to the nucleus in prophase, its activity is also increased relative to Polo[WT]-GFP. Using targeted point mutagenesis, we could not obtain NLS mutants that disrupt the nuclear localization of Polo without disrupting the PBD–KD interaction (K33E/K38E/K106A/R117A, K106D/K107D, and K106D/K107D/R117A) were tested. However, the T182D mutation makes Polo constitutively active without preventing nuclear localization. Expression of Polo[T182D]-GFP also fails to rescue hatching and allows only partial rescue of head development. These results suggest that Polo inactivation by dephosphorylation in the activation loop and PBD–KD reciprocal inhibition is required for Polo function in vivo.

To test if disruption of Polo regulation mechanisms causes dominant effects, we tested the expression of the different forms of Polo-GFP in the presence of endogenous Polo (Supplementary Fig. 8a–c). Expression of Polo[WT]-GFP has no detectable impact on head development (Fig. 6c). Similarly, expression of the kinase-dead, the PBD mutant, or the T182A mutants has no effect. By contrast, expression of the T182D or the NLS7A mutants, both associated with constitutive kinase activity, causes obvious developmental defects characterized by smaller and rougher eyes in approximately half the flies. These defects are not accompanied by a loss of viability as the vast majority of flies expressing all forms of Polo-GFP hatch (Supplementary Fig. 8d).

Finally, we examined the effects of the expression of different forms of Polo in the embryo. We found that Polo[T182D]-GFP and Polo[NLS7A]-GFP are toxic, causing a majority of embryos to fail development before hatching, while the loss-of-function mutants have little effects and Polo[WT]-GFP has an intermediate effect on hatching (Fig. 6d). All forms of Polo-GFP were expressed at similar levels, a few fold higher than endogenous Polo (Fig. 6e). Defects in hatching are mirrored by higher levels of mitotic MPM2 phospho-epitopes, known to depend on Polo activity[4, 39] (Fig. 6e). We used IF in 0–2 h-old embryos to analyze their phenotypes at the syncytial stage. We found that after fixation, the GFP was not visible and we were also unable to detect it with an antibody. Nevertheless, we stained embryos for tubulins α and γ, Lamin B, and DAPI. Embryos expressing Polo[T182D]-GFP and Polo[NLS7A]-GFP displayed the highest proportions of major developmental defects at the syncytial mitoses stage characterized by abnormal or asynchronous nuclei, free centrosomes, and large gaps in nuclear distribution (Fig. 6f, g; Supplementary Fig. 9). Embryos expressing Polo[WT]-GFP showed a higher proportion of minor developmental defects including anaphase bridges and free centrosomes that may not lead to lethality. Interestingly, expression of Polo[T182D]-GFP or even Polo[WT]-GFP, but not Polo[NLS7A]-GFP, caused a fraction of eggs to arrest in metaphase of meiosis I, suggesting that a gain of Polo activity in the oocyte nucleus may interfere with meiotic progression. Altogether, these results indicate that constitutive kinase activity of Polo is

**Fig. 4** Nuclear activity of Polo activates Cdc25 in prophase. **a** GFP-Cdc25 is rapidly excluded from the nucleus in prophase. The green box indicates the time of initial detection of GFP-Cdc25 in the cytoplasm before NEB set as $T_0$. Bar: 5 μm. In this experiment, cells were mock-treated with medium containing DMSO as a control for Supplementary Fig. 6b. **b, c** The nuclear exclusion of GFP-Cdc25 coincides with the nuclear import of Polo-RFP. Bar: 5 μm. (n = 10, error bars: SD). **d** Cells were transfected and treated with BI 2536 as indicated and were fractionated into cytoplasmic (C) and nuclear (N) fractions analyzed by western blots. The position of Cdc25 and its phosphorylated forms are indicated by arrows. MEK and Histone H3 are, respectively, cytoplasmic and nuclear proteins probed as controls. **e** Cdc25 phosphorylation is regulated by Polo kinase in the nucleus. D-Mel2 cells were transiently transfected as indicated. BI 2536 was added at 300 nM. Lysates were analyzed by western blotting. Inhibitory phosphorylation of Cdk1 was monitored using anti-pY15 antibodies. **f** Polo promotes Cdc25 activation in the nucleus. After transfection with the indicated constructs, Myc-Cdc25 (WT or phosphatase-dead) was immunoprecipitated and tested in a phosphatase assay. A representative experiment is shown with duplicate readings. AU arbitrary units. Error bars: SD. **g** The interaction between Polo and Cdc25 is regulated by Cdk1. Cells were transfected with Myc-Cdc25 and Polo-GFP (WT or NLS7A) plasmids, and treated or not with the Cdk1 inhibitor RO 3306 (10 μM). Cell lysates were subjected to immunoprecipitation with anti-GFP antibodies and immunoblotted with the indicated antibodies

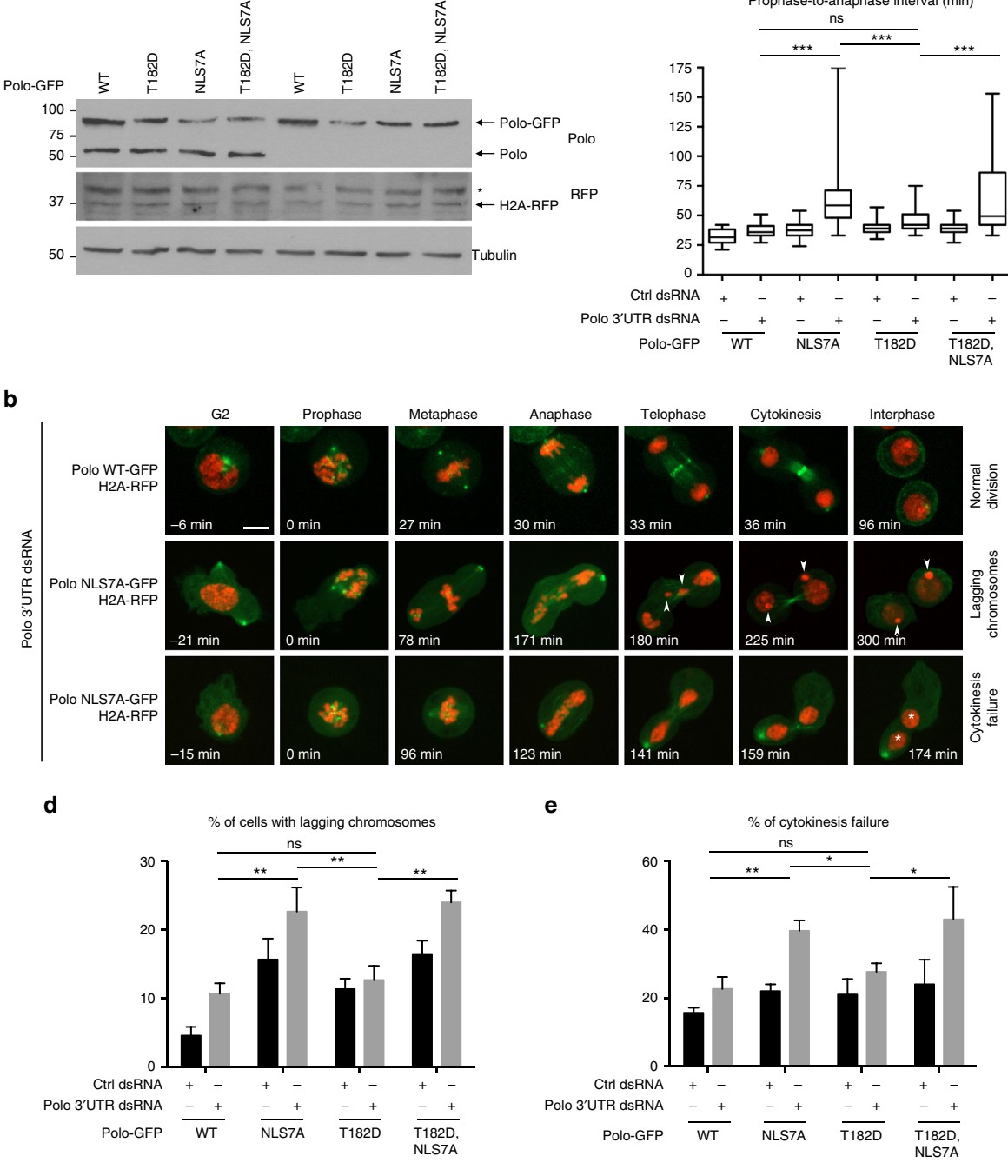

**Fig. 5** Failure of Polo to localize to the nucleus in prophase results in multiple mitotic defects. **a** In stable cell lines expressing H2A-RFP, expression of Polo-GFP (WT or mutants as indicated) was induced and cells were transfected with Polo 3′ UTR dsRNA (or control dsRNA). The next day, protein extracts were analyzed by western blots. *Non-specific band. **b** Cells from **a** were filmed. Arrowheads show lagging chromosomes. Asterisks indicate nuclei in a binucleated cell. Bar: 5 μm. **c** The time between prophase (visible onset of chromosome condensation, $T_O$) and anaphase was measured. **d** Quantification of cells with lagging chromosomes during division. **e** Quantification of cells showing a cytokinesis failure. **c**–**e** At least 40 cell divisions were scored for each condition in three independent experiments. Error bars: SD (*$P < 0.05$; **$P < 0.01$; ***$P < 0.001$; Student's $t$ test; ns non-significant)

detrimental during development, imposing requirements for nuclear exclusion in interphase and for interdomain inhibition of Polo via its NLS motif.

## Discussion

In this study, we have uncovered a mechanism that triggers Polo relocalization to the nucleus following its activation in prophase.

This mechanism is required for the regulation of Cdc25 and for successful mitosis. Our work functionally tests the contributions of Polo regulatory mechanisms in vivo.

We found that phosphorylation of Polo at its activation loop is required for its nuclear localization in prophase. The NLS in Polo that mediates nuclear import is located in the KD, at the interface with the PBD. Previous studies have suggested that phosphorylation of the activation loop induces the dissociation between the

KD and PBD[15, 18]. Our results from various approaches support this concept, but indicate that phosphomimetic mutation of the activation loop weakens but does not abolish the interaction between the KD and the PBD. It remains possible that actual phosphorylation completely disrupts the interaction. Nevertheless, the nuclear localization of Polo does not depend on kinase activity indicates that phosphorylation of the activation loop directly induces a structural change in Polo that exposes the NLS (Fig. 3g).

The available crystal structure of a complex between the KD and PBD of zebrafish Plk1 suggests that the PBD inhibits the KD

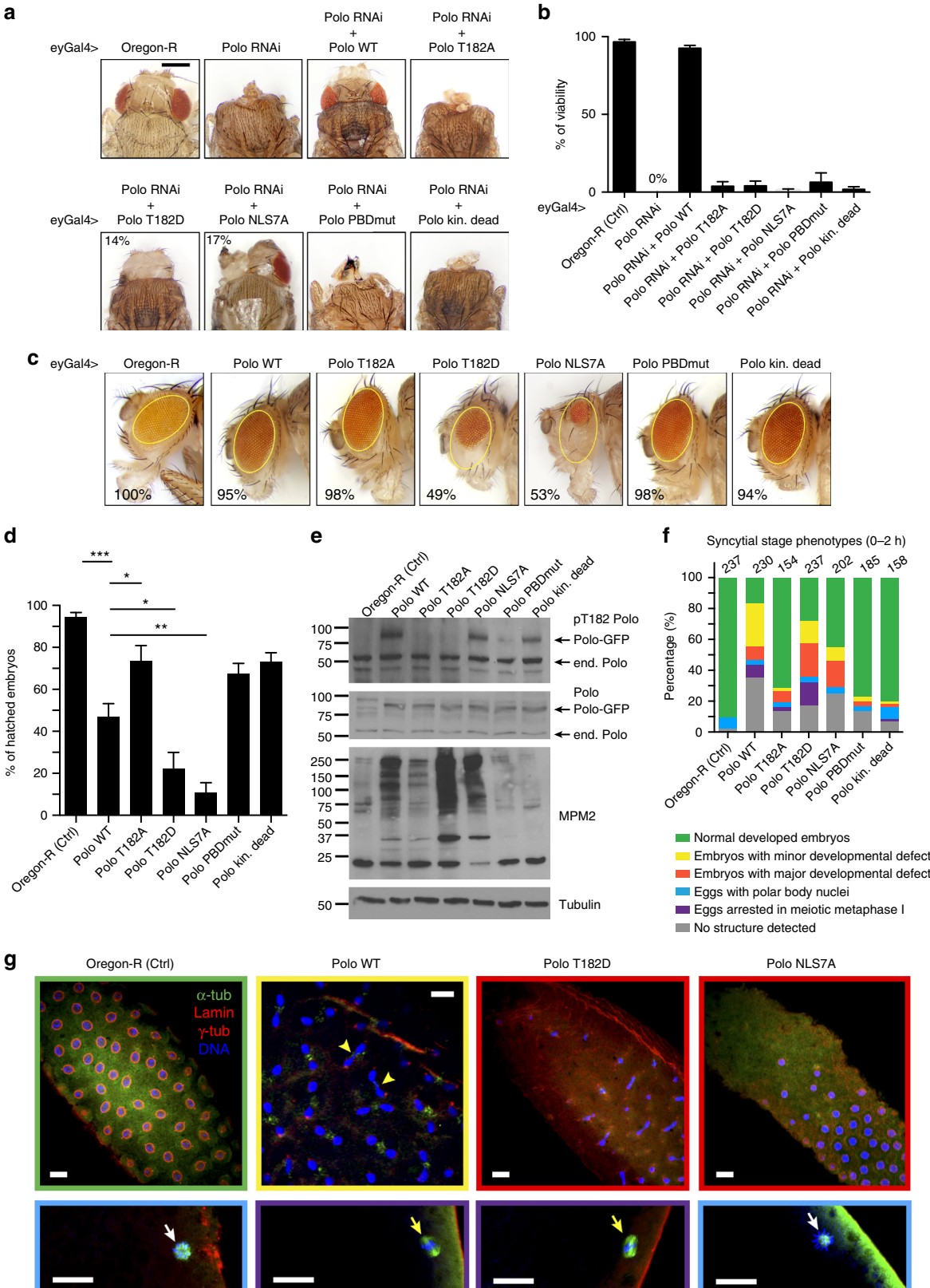

by restricting its conformational dynamics[17]. However, it does not reveal how activation loop phosphorylation could induce rearrangement of the interdomain contact and this effect may require the presence of the IDL, which is lacking in the crystallized complex. The same study showed that the IDL contributes to KD inhibition and hampers activation loop phosphorylation by Aurora A[17]. Because of its functional importance, the IDL was included in the KD constructions that we used in GST pulldown and BRET experiments. Alternatively, it remains formally possible that phosphorylation of the activation loop in Polo promotes an interaction with another protein required for the structural change in Polo that exposes the NLS. Undoubtedly, a crystal structure of full-length Polo$^{T182D}$ or a vertebrate equivalent would be illuminating. Interestingly, the fact that mutations abrogating the interdomain interaction in Polo increase catalytic activity to levels similar to the T182D mutation suggests that the only role of activation loop phosphorylation is to relieve KD inhibition by the PBD. This is in contrast from other kinases in which phosphorylation of the activation loop is necessary for full kinase activation[40, 41].

Our results show that activation loop phosphorylation of Polo is required for the nuclear accumulation of Polo, but suggest that it is not sufficient for this event in cell culture. Although the T182D mutation advances nuclear accumulation of Polo, it does not make Polo constitutively nuclear. It is possible that the T182D substitution does not perfectly mimic phosphorylation at Thr182, as this has been shown for the T210D mutation in human Plk1[42]. Alternatively, Polo could remain significantly retained in the cytoplasm by Map205 until the phosphorylation of Map205 at its CDK site disrupts its interaction with Polo[19]. On the other hand, Polo$^{WT}$-GFP is constitutively enriched in the nucleus in embryos. Using a phospho-specific antibody, we can detect pT182-Polo at all cell cycle stages in embryos[43]. Map205 phosphorylation by Cdk1 may be similarly more constitutive in the rapid embryonic cell cycles than in dividing cells in culture.

We showed that the NLS in Polo is subject to regulated intramolecular masking. Moreover, NLS residues play a dual role: nuclear localization during prophase and inhibition of activity via interdomain interaction during interphase. The NLS motifs in transcriptional regulators including NF-AT4, STAT1, and b-Myb are masked intramolecularly and exposed upon structural changes induced by post-translational modifications[44, 45]. However, in these cases, the NLS was not shown to participate in the regulation of intrinsic protein activity as we have shown for Polo. It would be interesting to explore if other proteins use similar mechanisms to coordinate activity and nucleocytoplasmic localization.

Based on our findings, we propose a model for how the spatiotemporal regulation of Polo contributes to mitotic entry (Fig. 7). Once activated, Polo localizes to the nucleus, triggering the relocalization of Cdc25 from the nucleus to the cytoplasm and

its activation. In the cytoplasm, Cdc25 activates Cyclin B-Cdk1, which promotes its own nuclear import[46]. In the nucleus, Cyclin B-Cdk1 phosphorylates Cdc25 to promote its binding and phosphorylation by Polo. This mechanism appears as a logical link in the Cyclin B-Cdk1 auto-amplification process making mitotic entry a switch-like transition. We showed that failure of Polo to localize in the nucleus in prophase is associated with defects in chromosome segregation and subsequent cytokinesis. We speculate that these defects could result from insufficient or untimely Cyclin B-Cdk1 activation, and a resulting failure in phosphorylation of important substrates for chromosome condensation, kinetochore function, or spindle function.

In human cells, Plk1 activation depends on Aurora A kinase and occurs in the cytoplasm, presumably at centrosomes[24, 29]. Sequestering Plk1 to the nucleus prevents its activation[29]. In Drosophila, we previously showed that Polo activation is mediated largely by Aurora B[26]. In this study, we further showed that chemical inhibition of Aurora B delays accumulation of Polo in the nucleus in prophase. Phosphorylation of Polo by Aurora B is required for the detection of active Polo at centromeres, where it interacts with INCENP, the Aurora B activator, in mitosis[26]. We do not know if the phosphorylation of Polo by Aurora B in prophase occurs in the cytoplasm, the nucleus, or both. Although Polo is mostly cytoplasmic at steady state in interphase, it may already shuttle through the nucleus at a significant rate and be targeted there by Aurora B/INCENP in prophase, shifting the equilibrium to a more nuclear localization of Polo. Consistent with this idea, the nuclear localization of Polo-GFP in the embryo is not completely abolished by the T182A substitution when compared with the NLS7A mutation (Figs. 1a, 2e). It is also possible that phosphorylation of Polo by Aurora A at centrosomes triggers the initial increase in nuclear import of Polo, and that phosphorylation by Aurora B in the nucleus is required to maintain high levels of activated Polo. However, as Aurora B has known functions in both the nucleus and the cytoplasm, it is possible that initial phosphorylation of Polo by Aurora B in prophase simply occurs in the cytoplasm[47].

We have investigated how the coordination of Polo activation and localization helps its function toward the Cdk1-activating phosphatase Cdc25$^{String}$. In Drosophila, String is the only somatically expressed Cdc25 phosphatase[34, 48]. We document for the first time the spatiotemporal regulation of Cdc25$^{String}$ in the cell cycle. The nuclear exclusion of Cdc25 in prophase coincides with and requires the nuclear import of Polo. This localization dynamics of Cdc25 is reminiscent of that observed for Greatwall[49]. Interestingly, both Cdc25 and Greatwall depend on Polo and Cdk1 for their phosphorylation and nuclear exclusion, suggesting that they may be regulated by similar mechanisms[43, 49].

In humans, there are three Cdc25 homologs. Cdc25C and Cdc25B activate Cdk1 at the G2/M transition[33, 50]. Both

**Fig. 6** Coordination of Polo activity and localization is essential for its functions in vivo. **a** The eyeless-Gal4 (eyGal4) driver was used to induce simultaneous depletion of endogenous Polo and expression of different RNAi-insensitive forms of Polo in the developing head. Pharate adults were dissected out of pupae cases. Representative images are shown for each genotype. Bar: 0.5 mm. See Supplementary Fig. 7 for crosses and PCR validation. **b** The ability of RNAi-insensitive forms of Polo-GFP (WT or mutants) to rescue head development scored by percentages of viability (pupae hatching). Percentages represent at least 300 animals in three independent experiments. Error bars: SD. **c** Images of adult flies after overexpression of Polo-GFP (WT or mutants) in the presence of endogenous Polo. Bar: 0.5 mm. **d** Different forms of Polo-GFP were expressed in the female germline from transgenes driven by matα4-GAL-VP16. The percentage of hatched embryos laid by these females was scored. Results from three independent experiments were combined (n = 300). Error bars: SD (*P < 0.05; **P < 0.01; ***P < 0.001; Student's t test). **e** A gain of Polo function in embryos causes an accumulation of mitotic MPM2 epitopes. Extracts from 0–2 h embryos were probed by western blot as indicated. **f, g** A gain of Polo function causes developmental defects in meiosis and syncytial mitosis. Eggs and embryos laid by females expressing different forms of Polo-GFP were collected every 2 h and phenotypes were examined by immunofluorescence. After fixation, the GFP was not visible. For each genotype, results are compiled from 3 to 4 independent collections. The different categories are color-coded. In a fraction of eggs/embryos overexpressing Polo-GFP variants, we could not detect defined structures, suggesting that nuclei degenerated and/or remained deep where antibodies have poor access (gray category). Yellow arrowheads: anaphase defects. White arrows: polar body indicated normally completed meiosis. Yellow arrows: meiosis spindles blocked in metaphase I. Scale bars: 20 μm

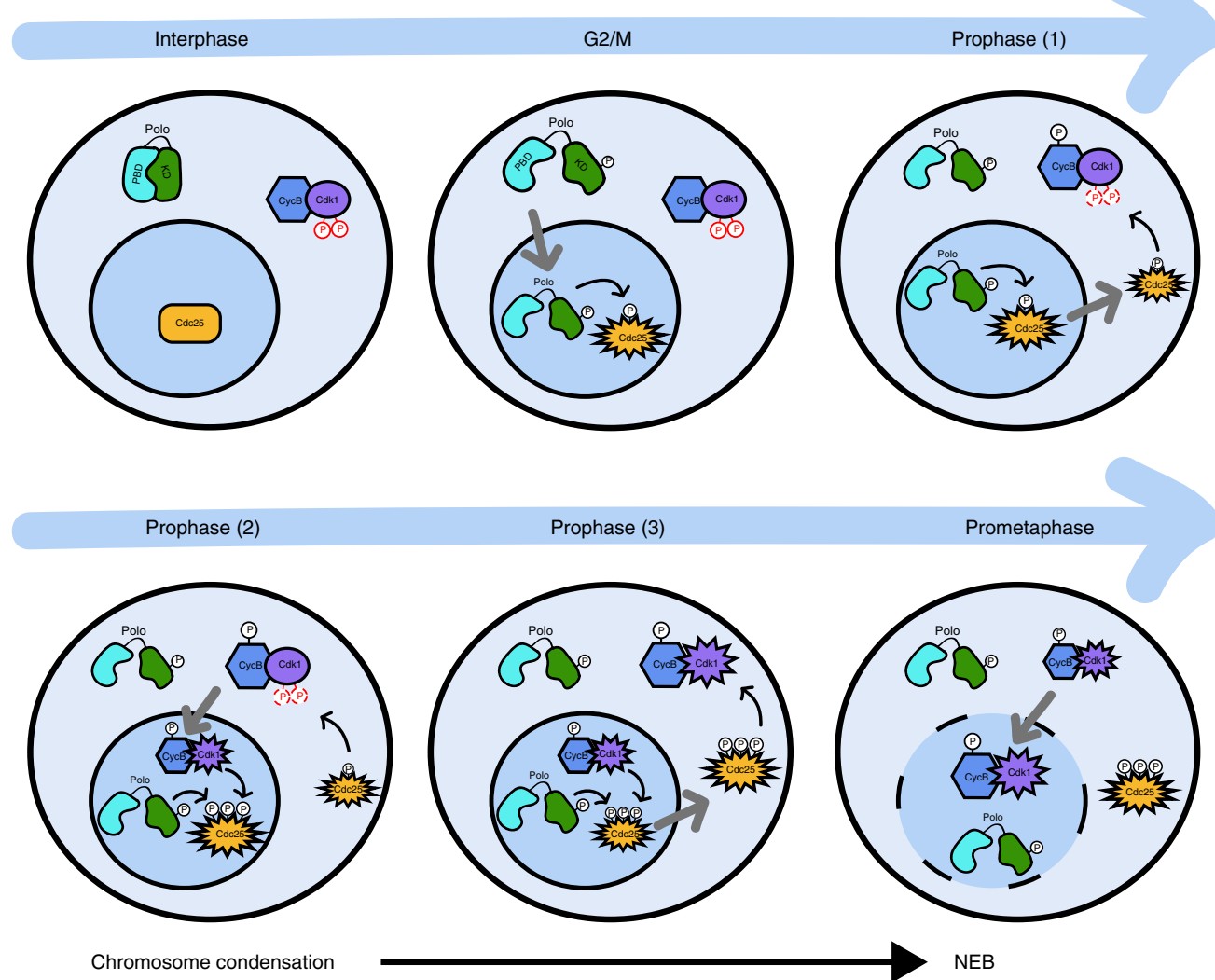

**Fig. 7** Spatiotemporal model for the coordination of mitotic entry. At the G2/M transition, activated Polo translocates to the nucleus and phosphorylates Cdc25. Cdc25 starts to relocalize to the cytoplasm during prophase where it begins to activate Cdk1 (1). In the nucleus, Cyclin B-Cdk1 reinforces Polo-dependent relocalization of Cdc25 (2). This positive feedback mechanism accelerates the generation of fully active Cyclin B-Cdk1 (3) allowing chromosome condensation and lamina disassembly at mitotic entry

phosphatases have been shown to be imported to the nucleus in prophase in a Plk1-dependent manner[51, 52]. Thus, the spatial regulation of Cdc25C and Cdc25B by Plk1 appears to be opposite from the regulation of Cdc25[String] by Polo. However, several pools of Cdc25C co-exist that exhibit distinct localizations and may be regulated differentially[53]. Reports on the regulation of Cdc25B are conflicting, suggesting that it promotes mitotic entry in the nucleus or in the cytoplasm, and this may be due to the existence of several Cdc25B splice variants[52, 54, 55]. Finally, Cdc25A, which is primarily implicated in the activation of cyclin-CDK enzymes at the G1/S transition, has been shown to localize mainly to the nucleus[33, 56, 57]. Common to all three human Cdc25 enzymes is that they shuttle between the nucleus and cytoplasm. Specificities in their spatiotemporal regulation may reflect the evolution of specialized functions. As String is the only Cdc25 expressed in somatic cells in *Drosophila*, its regulation by Polo uncovered here may reflect the most crucial wiring in basic cell cycle regulation shared by animals. Our results confirm that Polo activates Cdc25[String] in *Drosophila*, which is consistent with the canonical mechanism originally identified in Xenopus oocytes[22].

Importantly, this study dissects the importance of Polo regulatory modes in vivo, using embryos and the developing head tissues. The in vivo requirements of kinase activity, activating phosphorylation, and PBD phosphotarget binding activity provide additional biological support for the established basic model for PLK function[5, 13]. In addition, we uncovered a strong requirement for Polo interdomain inhibition, revealed by the dominant effects of the NLS and T182D mutants and their failure to rescue development from the loss of endogenous Polo. It would be interesting to explore which molecular mechanism is disrupted by constitutive Polo activity in cell cycle progression or other cellular functions. The fact that our point mutations in the NLS could not separate nuclear localization from interdomain inhibition further highlights the interlocking between the two functions of the NLS motif.

The novel mechanism coupling Polo activity and localization, and its functional importance during mitotic entry, add a previously missing centerpiece in the construction of a complete understanding of the spatiotemporal regulation of the cell cycle.

## Methods

**Plasmids and mutagenesis**. The plasmids were generated by Gateway recombination (Invitrogen). Coding sequences were first cloned into the pDONOR221 entry vector. They were then recombined into the relevant destination vectors for expression from copper-inducible (pMT) or constitutive (pAC5) promoters. The following expression vectors were generated: pMT-Polo$^{WT}$-GFP, pMT-Polo$^{T182A}$-GFP, pMT-Polo$^{T182D}$-GFP, pMT-Polo$^{NLS7A}$-GFP, pMT-Polo$^{T182A/NLS7A}$-GFP, pMT-Polo$^{T182D/NLS7A}$-GFP, pMT-Polo$^{K54M}$-GFP, pMT-Polo$^{P384R}$-GFP, pMT-Polo$^{M527R}$-GFP, pMT-Polo$^{P384R/M527R}$-GFP, pMT-NLS-Polo$^{WT}$-GFP, pMT-NLS-Polo$^{NLS7A}$-GFP, pAC5-Flag-KD + IDL$^{WT}$ (amino acids 1–349), pAC5-Flag-KD + IDL$^{T182A}$, pAC5-Flag-KD + IDL$^{T182D}$, pAC5-Flag-KD + IDL$^{NLS7A}$, pAC5-Flag-KD + IDL$^{T182A/NLS7A}$, pAC5-Flag-KD + IDL$^{T182D/NLS7A}$, pAC5-PrA-Map205, pMT-RFP-Lamin, pAC5-RFP-Lamin, pMT-H2A-RFP, pAC5-Myc-Cdc25(String), pMT-GFP-Cdc25(String).

The different GST-PBD (WT, P384R, M527R, P384R/M527R) (amino acids 350–576) plasmids were constructed similarly into the Gateway pDEST15 vector.

Amino-acid substitution mutants were generated using QuikChange Lightning Site-Directed Mutagenesis Kit (Agilent) following the manufacturer's protocol.

**Drosophila cell culture and drug treatments**. All cells were in the D-Mel2 background and were cultured in Express Five medium (Invitrogen) supplemented with glutamine, penicillin, and streptomycin. All stable cell lines were selected in medium containing 20 µg mL$^{-1}$ blasticidin. Expression of the copper-inducible transgenes were induced with CuSO$_4$ (300 µM) for at least 8 h. For Aurora B inhibition, cells were treated with 20 µM Binucleine 2 (EMD Millipore) for 1 h before being processed for time-lapse microscopy. For Polo kinase inhibition, cells were treated with BI 2536 (MedChem Express) for 1 h before being processed for time-lapse microscopy or IF. We used different concentrations of BI 2536 depending of the experiment (30 nM, 100 nM, 200 nM, 1 µM). For Cdk1 inhibition, cells were treated with 10 µM RO 3306 (Tocris Bioscience) for 1 h before being lysed.

**Transient transfections**. Transfections of D-Mel2 cells with plasmids were performed using X-tremeGENE HP DNA Transfection Reagent (Roche) following the manufacturer's instructions. For RNA interference, cells were transfected in six-well plates with 30 µg of Polo 3′ UTR double-stranded RNA (dsRNA) using Transfast reagent (Promega). The control dsRNA was generated against the sequence of the bacterial kanamycin resistance gene. Cells were analyzed 24 h later by immunoblotting or live cell imaging.

**Western blotting and immunofluorescence**. All uncropped images of western blots (WB) are presented at the end of the Supplementary information. Primary antibodies used in western blotting and IF were anti-Flag M2 produced in mouse (#F1804, Sigma, at 1:2000 dilution for WB), anti-GST from rabbit (#2622, Cell Signaling Technology, at 1:1000 dilution for WB), anti-GFP from rabbit (#A6455, Invitrogen, at 1:5000 dilution for WB), anti-GFP from mouse (#1218, Abcam, at 1:5000 dilution for WB), peroxidase-conjugated ChromPure rabbit IgG (#011-030-003, for PrA detection, Jackson ImmunoResearch, at 1:3000 dilution for WB), anti-α-tubulin DM1A from mouse (#T6199, Sigma, at 1:10,000 dilution for WB, at 1:1000 dilution for IF), anti-Myc 9E10 from mouse (#sc-40, Santa Cruz Biotechnology, Inc., at 1:2000 dilution for WB, at 1:500 dilution for IF), mouse monoclonal anti-Polo MA294 (a gift of D. Glover, University of Cambridge, Cambridge, UK, at 1:300 dilution for WB), anti-RFP from rabbit (#62341, Abcam, at 1:1000 dilution for WB), anti-Lamin Dm0 (DSHB Hybridoma Product ADL84.12, ADL84.12 was deposited to the DSHB by Fisher, P. A., at 1:100 dilution for IF), anti-P-cdc2 (Y15) from rabbit (#9111, Cell Signaling, at 1:2000 dilution for WB), anti-Cdk1 PSTAIR from mouse (#10345, Abcam, at 1:3000 dilution for WB), anti-phospho-Histone H3 from rabbit (#06-570, Millipore, at 1:300 dilution for IF), anti-α-tubulin YL1/2 from rat (#6160, Abcam, at 1:100 dilution for IF), anti-MEK from rabbit (#9122, NEB, at 1:2000 dilution for WB), anti-Histone H3 from rabbit (#9717, NEB, at 1:1000 dilution for WB), anti-γ-Tubulin GTU-88 from mouse (#T5326, Sigma, at 1:100 dilution for IF), and anti-MPM2 from mouse (from D. Glover, at 1:1000 dilution for WB).

For IF, cells were fixed with 4% formaldehyde during 10 min or with cold methanol (−20 °C) for 3 min. Cells were permeabilized and blocked in PBS containing 0.1% Triton X-100 and 1% BSA (PBSTB). Cells were incubated with primary antibodies diluted in PBSTB for 1 h at RT, washed three times in PBS and incubated with secondary antibodies diluted in PBSTB for 1 h at RT. Cells were washed three times in PBS before being mounted in Vectashield medium with DAPI (Vector Laboratories).

For IF in embryos, females were allowed to lay eggs on grape juice agar at 25 °C. Embryos aged 0–2 h were then collected, washed with PBS, dechorionated using a 50% bleach solution for 2 min and fixed for 5 min in a mixture of heptane: 33% formaldehyde in PBS (1:1). Vitelline membranes were then removed by repetitive methanol washes before rehydration with a 1:1 methanol/PBS solution and then with PBS containing 0.1% Tween 20 (PBT). Embryos were blocked for 30 min in PBT + 1% BSA before being incubated overnight at 4 °C with primary antibodies in PBT + 1% BSA. After washed in PBT, a 2-h incubation was done at room temperature with secondary antibodies (coupled to Alexa-488 (Invitrogen) or Cy3

(Jackson), at 1:200 dilution). After final washes, DAPI was used to stain DNA. Embryos were mounted in Vectashield medium (Vector Laboratories). Images were taken using a confocal Laser Scanning Microscope LSM700 (Zeiss), using a ×63 oil objective.

**GST-pulldown assay**. Pelleted cells transfected with Flag-KD + IDL (WT or mutants) from confluent 25-cm$^2$ flasks were lysed in 50 mM Tris-HCl pH 7.5, 150 mM NaCl, 1 mM EDTA, 10% glycerol, 0.2% Triton X-100, 1 mM PMSF, 10 µg mL$^{-1}$ aprotinin and 10 µg mL$^{-1}$ leupeptin, and lysates were centrifuged during 10 min at 4 °C. Clarified lysates were incubated with GST–PBD (WT or mutants) or GST sepharose beads during 2 h at 4 °C. Beads were washed five times with lysis buffer before SDS–PAGE and immunoblotting.

**Protein affinity purification**. For Protein A affinity purifications, transfected cells with PrA-Map205 (from confluent 25-cm$^2$ flasks) were harvested and resuspended in 400 µL of lysis buffer supplemented with protease inhibitors. Lysates were clarified by centrifugation at 19,000×$g$ for 10 min in a tabletop centrifuge at 4 °C. Supernatants were incubated for 1 h at 4 °C with 20 µL of DynaBeads (Invitrogen) that we previously conjugated to rabbit IgG. Beads were washed five times with 1 mL of lysis buffer for 5 min at 4 °C. Purification products were eluted by heating at 95 °C for 5 min in 20 µL of Laemmli buffer 2× (Sigma-Aldrich) and analyzed by western blotting.

**Immunoprecipitation and kinase assay**. For immunoprecipitation of Polo-GFP (WT or mutants), extracts were prepared as above and lysates were incubated with anti-GFP antibodies (#A6455, Invitrogen) for 1 h at 4 °C and then incubated with 20 µL of Protein A-conjugated Dynabeads (Life Technologies) for 45 min at 4 °C, before being washed in lysis buffer as above.

For kinase assays, reactions were performed in kinase buffer (20 mM K-HEPES pH 7.5, 2 mM MgCl$_2$, 1 mM DTT) with 0.5 µM ATP,$^{32}$P-γ-ATP, and 1 µg casein at 30 °C for 15 min with agitation. For Polo inhibition in the kinase assay, BI 2536 was added at 300 nM. Reactions were stopped with the addition of the Laemmli buffer and heating at 95 °C for 2 min. Samples were separated by SDS–PAGE and transferred onto nitrocellulose membranes for autoradiography and western blotting with anti-GFP antibodies (#1218, Abcam).

**Colorimetric phosphatase assay**. Our protocol was adapted from a published protocol[58] (see paragraph: basic protocol 2). In brief, immunoprecipitated Myc-Cdc25 beads were incubated for 1 h at room temperature in a 96-well plate with 200 µM of Cdk1 peptides phosphorylated at Tyr15 (EKIEKIGEGTpYGVVYKGR NRL, Bio Basic Inc.) diluted in a colorimetric buffer (20 mM Tris pH 7.5, 5 mM MgCl$_2$, 1 mM EGTA, 0.02% β-mercaptoethanol, and 0.1 mg mL$^{-1}$ BSA). The reactions were stopped by addition of 0.09 M HClO$_4$ and mixed with the malachite green reagent, which emits at 620 nm upon binding of free phosphate. Samples were analyzed on a colorimetric plate reader. A phosphate standard curve using serial dilutions of 1 mM phosphate solution (KH$_2$PO$_4$) and a blank sample containing no phosphatase were used as controls.

**Subcellular fractionations**. Cytoplasmic and nuclear extracts were obtained using the NE-PER Nuclear and Cytoplasmic Extraction Reagents Kit (#78833) according to the manufacturer's instructions (Thermo Scientific).

**Microscopy**. Images of fixed cells were acquired on an AxioImager microscope (Carl Zeiss) with a ×100 oil objective (NA 1.4) and an AxioCam HRm camera (Carl Zeiss), using AxioVision software (Carl Zeiss).

Live imaging was performed using a Spinning-Disk confocal system (Yokogawa CSU-X1 5000) mounted on a fluorescence microscope (Zeiss Axio Observer.Z1) using an Axiocam 506 mono camera (Zeiss) and a ×63 oil objective (NA 1.4).

For time-lapse microscopy of D-Mel2 cells, cells in culture were plated in a Lab-Tek II chambered coverglass (#155409, Thermo Fisher Scientific). The GFP fluorescence signal in the nucleus and in the whole cell was quantified directly with the Zen software. To measure the GFP fluorescence ratio (nucleus/total cell), time-lapse images were collected at 3 min intervals during 66 min in the period preceding the NEB. For each time-lapse, an outline was drawn in a single in-focus plane around the plasma membrane (total cell) and around the nuclear envelope (nucleus). For each selected region (total cell or nucleus), GFP intensity was measured (area × average signal per pixel in the selected region). The GFP fluorescence ratios (nucleus/total cell) were calculated and plotted on a graph.

For live analysis of Drosophila syncytial embryos, 0–3-h-old embryos were first dechorionated in 50% bleach, aligned on a coverslip (#P35G-1.5-14-C, MatTek) and covered with halocarbon oil. Nineteen confocal sections of 1 µm were collected per time point for each embryo. Images were treated using Zen software (Zeiss) and Fiji software (NIH). The GFP fluorescence intensity ratio in prophase (nucleus/cytoplasm) per embryo was calculated at a single $z$-plane (the plane with the best focus) with the Zen software by dividing the GFP intensity signal of 20 nuclei per embryo by the GFP intensity signal of the same size region in the cytoplasm. Scatter dot plots and statistical analysis were performed using GraphPad Prism 6. For embryo pictures, deconvolution was carried out using the

Fiji software and "Diffraction PSF 3D" and "Iterative Deconvolution 3D" plugins. The number of iteration for deconvolution was set to 3. A final projection was made on the z-planes containing the nucleus and the final images were cropped to 138 × 138 pixels. All images were prepared for publication using Adobe Photoshop to adjust contrast and brightness.

**Fly genetics**. Fly husbandry was conducted according to standard procedures. All crosses were performed at 25 or 27 °C. The WT strain used was Oregon R. Transgenic flies for expression of UAS-Polo-GFP (WT and mutants) were created by site-directed insertions after injections of our plasmids (pUAS-K10attB-based) in the attP154 strain by BestGene. Insertions on the third chromosome allowing very similar expression levels were selected for experiments (PhiC31 integrase-mediated site-specific transgenesis, BestGene Inc). The UAS-Polo RNAi strain used for depletion of Polo was obtained from Vienna *Drosophila* Resource Center (#20177). Expression of Polo transgenes in the early embryo was driven by matα4-GAL-VP16 (#7062, Bloomington *Drosophila* Stock Center). Expression of Polo transgenes and depletion of endogenous Polo in the developing head tissues was driven by eyeless-Gal4 (#5534, Bloomington *Drosophila* Stock Center).

For fertility tests, well-fed females were mixed with males in tubes containing grape juice agar and allowed to lay eggs for 1 day before being removed. The percentage of hatched embryos was counted 24 h later.

For viability tests, flies were crossed and the number of observed flies (number of pupae hatching) relative to their expected number in the progeny (total number of pupae) was expressed as a percentage.

**Single fly genomic DNA extraction and PCR validation**. The single fly is placed in a PCR tube, resuspended in 50 μL of "squish buffer" (10 mM Tris-HCl pH 8, 1 mM EDTA, 25 mM NaCl and 0.2 mg mL$^{-1}$ Proteinase K) and smashed against the wall of the tube. The PCR tube is placed in the thermocycler with the following program: 37 °C for 30 min (digestion) and 95 °C for 2 min (heat-kill enzyme). The tube is centrifuged for 7 min at 19,000×g and the supernatant (genomic DNA) is transferred to a new tube.

To verify the presence of each transgene in a fly, PCR reactions were performed using 1 μL of genomic DNA, 0.5 μL of each primer (100 μM), 23 μL H$_2$O nuclease free, and 25 μL of PCR Master Mix 2X (Promega) following the manufacturer's protocol.

The primers used for GFP sequence amplification (455 bp) were forward: 5′-AAGGGCGAGGAGCTGTTCAC-3′ and reverse: 5′-GGCCATGATATAGACG TTGTGGCTGTT-3′.

The primers used for Gal4 sequence amplification (520 bp) were forward: 5′-ATGAAGCTACTGTCTTCTATCGAACAAGCATG-3′ and reverse: 5′-AATCA AATCCATGAAGAGCATCCCTGGG-3′.

The primers used for Polo RNAi sequence amplification (600 bp) were forward: 5′-GAGGCGCTTCGTCTACGGAG-3′ and reverse: 5′-GATCGGGGATTTCGGG TTGGC-3′.

**Bioluminescence resonance energy transfer**. The KD–IDL (amino acids 1–349) portion of Polo N-terminally fused to RLucII (Luc-KD) and the PBD (amino acids 350–576) of Polo C-terminally fused to GFP10 (PBD–GFP) were expressed from pIRESHyg3 plasmids. A fragment of *Drosophila* Map205 (residues 254–416) with a C-terminal Myc tag (Map) was expressed from a pcDNA3.1 plasmid.

HEK293T cells were cultured in DMEM medium (Invitrogen) supplemented with 10% FBS (Invitrogen) and with penicillin/streptomycin (Wisent). For titration curves, 30,000 HEK293T cells were seeded in 96-well white plate (CulturePlate; PerkinElmer Inc.) coated with poly-L-ornithine and transfected the day after using Lipofectamine (Invitrogen). The amount of transfected Luc-KD plasmid (40 ng) and Map plasmid (200 ng) were keep constant, while the amount of PBD–GFP plasmid was increased between 0 and 400 ng. The empty pIRESHyg3 plasmid was used to complete 640 ng of total DNA transfected per well where needed. Lipofectamine was used in a 3:1 ratio (3 μL Lipofectamine: 1 μg DNA). Transfections for titration curves were done in triplicate. The BRET reaction was induced by the addition of coelenterazine 400a (Gold Technology) and measurements and analysis were performed as described[59] except that BRET2 was read using the Synergy NEO microplate reader (BioTek Instruments).

Raw data were analyzed using the Prism 5.00 (GraphPad) software. BRET$_{50}$ values correspond to the GFP10/RLucII ratio required to reach 50% of the maximum BRET signal and was obtained by the extrapolation of the titration curves using a one site binding hyperbolic fitting of the data. BRET$_{50}$ differences between the WT and mutants were assessed statistically using an F-test.

**Structure rendering**. Structure rendering was done using PyMOL 1.4. The structure of zebrafish Polo in complex with the inhibitory peptide from *Drosophila* Map205 (Protein Data Bank accession no. 4J7B[17]) was used to generate Figs. 2f, g, and 3g.

**Data availability**. The data sets generated during and/or analyzed during the current study are available from the corresponding author on reasonable request.

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

## Acknowledgements

We thank Jean-Claude Labbé, Matthew Smith, Lionel Pintard, and members of the Archambault lab for comments on the manuscript. We thank Myreille Larouche for help with the figures. This work was supported by grants from the Canadian Institutes of Health Research (CIHR) and from the Natural Sciences & Engineering Research Council (NSERC) to V.A. D.K. held a fellowship from the Cole Foundation. D.G., H.M. and V.A. hold fellowships from the Fonds de recherche du Québec—Santé (FRQS). IRIC is supported in part by the Canada Foundation for Innovation and the FRQS.

## Author contributions

D.K. and V.A. designed the project. D.K., D.G., H.M. and K.N. conducted the experiments. D.K., D.G., H.M., K.N. and V.A. analyzed the data. D.K., D.G., K.N., H.L. and V. A. wrote the paper.

## Additional information

**Competing interests:** The authors declare no competing financial interests.

