## [Peer Review File · Nature Communications]

Reviewers' comments:

Reviewer #1 (Remarks to the Author):

The manuscript by Kachaner et al demonstrates that Polo activation is coupled with its nuclear localization and investigates the functional importance of this mechanism. The authors have applied an impressive combination of cell biology and biochemical experiments to arrive at their conclusions. The strength and clarity of the findings presented in this manuscript rely on the careful analysis and interpretation of the T182A, T182D and NLS7A Polo mutants. The findings presented in this manuscript will be of interest to the field of researchers studying mitotic regulation. The manuscript is well written and the layout of the figures is clear and easy to follow.

1. To enhance this study, there are some key experiments that should be included to support their conclusions. The phenotypes of the NLS7A mutant appear to be more penetrant than either the T182A or T182D mutants (lagging chromosomes, cytokinesis, prophase to anaphase timing, hatching and eye development). In some cases, data for the T182A mutant was not shown. I am concerned that the NLS7A mutant is affecting more than just the nuclear localization of Polo. The NLS7A mutant has both increased kinase activity in vitro and delayed nuclear localization. The current set of mutants examined make it difficult to assess whether the added penetrance of the NLS7A mutant is due to the delayed nuclear localization and/or the increased kinase activity.

Ideally, a smaller set of NLS substitutions that prevent nuclear localization without increasing the kinase activity would be used. Given the interdomain dissociation model for Polo activation and the importance of the NLS residues in this interaction, it may not be possible to construct such a mutant. The T182A mutant has delayed nuclear accumulation but reduced activity in vitro while the T182D mutant has increased kinase activity in vitro but accelerated nuclear localization. The phenotypes for each of these mutants are milder than the NLS7A suggesting that both the delayed nuclear localization and higher activity are contributing to the penetrance of the NLS7A mutant. To address the concerns with the NLS7A mutant, the authors could add an NLS to the NLS7A mutant to restore or accelerate the nuclear localization of this mutant and assess the penetrance of the resulting phenotypes, which may be milder like the T182D mutant.

2. Figure 1D - Inhibition of Aurora B may have indirect effects on the nuclear localization of

Polo WT-GFP. The authors could provide more direct evidence by demonstrating that the nuclear localization of the T182D mutant is insensitive to Aurora B inhibition with Binucleine 2. It would also be beneficial to validate the inhibition of T182 phosphorylation on Polo WT-GFP treated with Binucleine 2.

3. The authors should acknowledge the work of Paschal et al Chromosoma (2012), which used gene targeting to replace the native PLK1 locus in human cells with either PLK1 T210A or PLK1 T210D (equivalent to T182 residue). In this setting, the T210D mutant did not have elevated kinase activity in vivo and failed to complement the kinase's essential role in chromosome congression.

4. It would be helpful if the authors could assess the activity of their mutants in vivo by monitoring the phosphorylation of known Polo substrates.

5. The authors state in the discussion that T182 phosphorylation can be detected with a phospho-specific antibody at all cell cycle stages in embryos. The authors should determine whether the NLS7A mutant can be phosphorylated at T182.

6. Figure 7 would benefit from a more detailed model that includes the regulation described in figures 1-3. The model should include the exposure of the NLS sequence by Aurora B dependent T182 phosphorylation of Polo (disruption of the kinase and the PBD domains).

Reviewer #2 (Remarks to the Author):

The manuscript from Kachaner et al., consists of many very well executed experiments focused on investigating the relevance of the NLS present in the key mitotic kinase, Polo.

It begins by documenting the localisation of various mutated versions of Polo-GFP, in *Drosophila* embryos and Dmel-2 cells, moves on to biochemically assess the relationship between the NLS and key phosphorylation events, relates it to cdc25 localisation and activity and finally looks at the some of the functional consequences of expressing mutated versions in the presence or absence of endogenous Polo.

My assessment is that, although on the surface, it doesn't seem to constitute enough of an advance to warrant publication in Nature Communications, it does have the potential to be of great interest to many researchers in the field of mitosis. *Drosophila* Polo has a long and illustrious history and understanding the complex relationships between the kinase's localisation and CDK1 / Cdc25 activation is undoubtedly important.

MAJOR POINTS:

1. The main disappointment with the study, as it currently stands, is the lack of functional analysis using the *Drosophila* embryo. The study begins by documenting the localisation of the different Polo-GFP mutants - Thr182A, Thr182D, NLS7A - driven in the embryo using the Mat-a4-GAL4 driver, in the presence of endogenous Polo. Then, at the very end of the study we are told that, for Thr182D and NLS7A, this leads to a failure to hatch of ~80% and ~90% of embryos. But there is nowhere an indication of what the consequences on mitosis, karyokinesis, cell cycle timing etc are in embryos. There is nice work in Dmel2 cells (Figure 5), but it does seem amiss to leave out an analysis in syncytial embryos. Similarly, the localisation of Polo-GFP NLS7A in embryos is fascinating - it appears to be excluded, post NEB, from centrosomes, kinetochores/centromeres and the spindle and instead localises to the endoplasmic reticulum (Figure 2E). Yet, this is skipped over in the text - all that is mentioned is "NLS7A abolished the nuclear localisation of Polo in prophase" (line 153-4). The authors must have considered what this might mean. In my view, it is imperative to complement the cell biology of Dmel2 cells presented in Figure 5 with an investigation of the functional consequences of expressing the mutated versions of Polo-GFP in the early embryo, correlating this to the sub-cellular localisations.

2. It is not clear from the text, Figure legend or Methods whether the structural figures in Fig 2F and G are the authors re-rendering of published structural data or the result of structural homology modeling of *Drosophila* Polo. Line 859 (Figure 2 legend) says, "structural rendering was performed using PyMOL 1.4 built-in commands", but that's all we're told. There is no mention in the Methods to help clarify.

3. It would be good to see the levels of the transgenes expressed in the various cells and tissues - especially in the attempted rescues of the *eyeless*-GAL4 RNAi in Figure 6. Although we are told in the Methods that "Insertions on the third chromosome allowing very similar expression levels 607 were selected for experiments", it is possible that the variable level of rescue seen after expressing some of the variants may, at least in part, be due to expression differences. It would help if this could be ruled out.

4. There is no information on the quantification of images.

E.g. Figure 1 C and D, Figure 2D, Figure 4 C - "GFP fluorescence ratio (nucleus/total cell)" - we are not told in Figure legend or Methods how these were calculated. This is important, especially as I'm not convinced that Binucleine 2 delays PoloWT-GFP entry into the nucleus in prophase, as stated in line 142. The data presented in Figure 1D shows only a single time-point as being significantly different between DMSO and Binucleine, suggesting only a minimal delay, if any. Is there any other, stronger data to support the author's conclusion?

MINOR POINTS:

1. In line 47/48, the authors say Polo becomes concentrated on centromeres at late prophase. Yet, in line 82, they say Polo appears on centromeres before NEB. They should be consistent.
2. *Drosophila* needs to be italicised throughout
3. Line 591 - "*drosophila*" needs to be "*Drosophila*", as well as italicised.
4. The "Microscopy" in the Methods needs to be more detailed - the use of deconvolution 3D software implied multiple z-images per time-point, but I could not find details of this.

Response to reviewer comments on Kachaner et al (ms no NCOMMS-17-10990)

We wish to thank the reviewers for their thoughtful comments, suggestions and criticisms. We have conducted several additional experiments and now present a much stronger manuscript. New results are presented in Figures 1D, 4F, 6E-G, S1C, S2C-E, S4B-D, S8C and S9. In the manuscript text, changes are in green. Please find our answers to all of the reviewers' points below, in blue.

Reviewer #1 (Remarks to the Author):

The manuscript by Kachaner et al demonstrates that Polo activation is coupled with its nuclear localization and investigates the functional importance of this mechanism. The authors have applied an impressive combination of cell biology and biochemical experiments to arrive at their conclusions. The strength and clarity of the findings presented in this manuscript rely on the careful analysis and interpretation of the T182A, T182D and NLS7A Polo mutants. The findings presented in this manuscript will be of interest to the field of researchers studying mitotic regulation. The manuscript is well written and the layout of the figures is clear and easy to follow.

We thank the reviewer for expressing such a positive opinion of our work.

1. To enhance this study, there are some key experiments that should be included to support their conclusions. The phenotypes of the NLS7A mutant appear to be more penetrant than either the T182A or T182D mutants (lagging chromosomes, cytokinesis, prophase to anaphase timing, hatching and eye development). In some cases, data for the T182A mutant was not shown. I am concerned that the NLS7A mutant is affecting more than just the nuclear localization of Polo. The NLS7A mutant has both increased kinase activity in vitro and delayed nuclear localization. The current set of mutants examined make it difficult to assess whether the added penetrance of the NLS7A mutant is due to the delayed nuclear localization and/or the increased kinase activity.

Ideally, a smaller set of NLS substitutions that prevent nuclear localization without increasing the kinase activity would be used. Given the interdomain dissociation model for Polo activation and the importance of the NLS residues in this interaction, it may not be possible to construct such a mutant. The T182A mutant has delayed nuclear accumulation but reduced activity in vitro while the T182D mutant has increased kinase activity in vitro but accelerated nuclear localization. The phenotypes for each of these mutants are milder than the NLS7A suggesting that both the delayed nuclear localization and higher activity are contributing to the penetrance of the NLS7A mutant. To address the concerns with the NLS7A mutant, the authors could add an NLS to the NLS7A mutant to restore or accelerate the nuclear localization of this mutant and assess the penetrance of the resulting phenotypes, which may be milder like the T182D mutant.

The reviewer is correct. We have also discussed these caveats in the text and have been careful in our interpretations. As we mentioned in the manuscript, we have tried hard to make mutants that prevent nuclear localization without increasing kinase activity. The idea was to inactivate the NLS while preserving the inter-domain contacts. We therefore mutated residues that did not appear to be involved in inter-domain salt bridges and other direct contacts but that contributed to the NLS. The mutants we made and tested are: 1) K33E/K38E/K106A/R117A and 2) K106D/K107D and 3) K106D/K107D/R117A. As anticipated by the reviewer, none of these mutants had the desired effect. As we mention in the text, this observation further emphasizes the close coupling between NLS function and inter-domain regulation in Polo.

We have done the experiment suggested by the reviewer. We have fused the NLS of Polo on its N-terminus. The resulting NLS-Polo^{WT}-GFP or NLS-Polo^{NLS7A}-GFP is constitutively enriched in the nucleus (new Fig S4C-D). Interestingly, cells stopped dividing following the expression of these fusion proteins. This result suggests that constitutively nuclear Polo is toxic, imposing a need for nuclear exclusion of Polo in interphase.

2. Figure 1D - Inhibition of Aurora B may have indirect effects on the nuclear localization of Polo WT-GFP. The authors could provide more direct evidence by demonstrating that the nuclear localization of the T182D mutant is insensitive to Aurora B inhibition with Binucleine 2. It would also be beneficial to validate the inhibition of T182 phosphorylation on Polo WT-GFP treated with Binucleine 2.

We have done this experiment. We found that inhibition of Aurora B with Binucleine 2 did not prevent the nuclear localization of Polo^{T182D}-GFP, as predicted (new Figs. 1D and S2). We also verified by Western blot that treatment with Binucleine 2 prevents phosphorylation of Polo^{WT}-GFP (new Fig S2E; see also Kachaner et al, 2014).

3. The authors should acknowledge the work of Paschal et al Chromosoma (2012), which used gene targeting to replace the native PLK1 locus in human cells with either PLK1 T210A or PLK1 T210D (equivalent to T182 residue). In this setting, the T210D mutant did not have elevated kinase activity in vivo and failed to complement the kinase's essential role in chromosome congression.

We have now mentioned this study in the discussion.

4. It would be helpful if the authors could assess the activity of their mutants in vivo by monitoring the phosphorylation of known Polo substrates.

There are few known phosphorylation substrates of Polo in Drosophila cell division, and no phosphospecific antibodies that recognize these substrates at Polo-phosphorylation sites. However, we have used the MPM-2 antibody, which recognizes Polo-dependent phosphoepitopes, to assess the global levels of Polo activity in extracts from embryos expressing different forms of Polo. MPM-2 reactivity has been shown to reflect Polo activity in Drosophila

(Logarinho & Sunkel, 1998; do Carmo Avides, 2001). We observed that a gain of Polo function in embryos strongly increased MPM-2 signals (new Fig. 6E).

In addition, we have further characterized the regulation of Cdc25^{String} by Polo. We had already shown in our first submission that Polo activity induces Cdc25 phosphorylation and relocalization in cells. We have now tested the effect of Polo on Cdc25 phosphatase activity. We found that a gain of Polo activity in the nucleus increases Cdc25 activity *in vitro* after immunoprecipitation (new Fig. 4F). This result reinforces the idea that Polo positively regulates the function Cdc25 in promoting mitotic entry.

5. The authors state in the discussion that T182 phosphorylation can be detected with a phospho-specific antibody at all cell cycle stages in embryos. The authors should determine whether the NLS7A mutant can be phosphorylated at T182.

We have now done this control in both embryos (new Fig. 6E) and in cells (new Fig. S2E). The NLS7A mutant can indeed be phosphorylated at T182.

6. Figure 7 would benefit from a more detailed model that includes the regulation described in figures 1-3. The model should include the exposure of the NLS sequence by Aurora B dependent T182 phosphorylation of Polo (disruption of the kinase and the PBD domains).

We thank the reviewer for this suggestion. We provide a new version of the model with these improvements (new Fig. 7)

Reviewer #2 (Remarks to the Author):

The manuscript from Kachaner et al., consists of many very well executed experiments focused on investigating the relevance of the NLS present in the key mitotic kinase, Polo.

It begins by documenting the localisation of various mutated versions of Polo-GFP, in *Drosophila* embryos and Dmel-2 cells, moves on to biochemically assess the relationship between the NLS and key phosphorylation events, relates it to cdc25 localisation and activity and finally looks at the some of the functional consequences of expressing mutated versions in the presence or absence of endogenous Polo.

My assessment is that, although on the surface, it doesn't seem to constitute enough of an advance to warrant publication in Nature Communications, it does have the potential to be of great interest to many researchers in the field of mitosis. *Drosophila* Polo has a long and illustrious history and understanding the complex relationships between the kinase's localisation and CDK1 / Cdc25 activation is undoubtedly important.

We are glad that this reviewer too is positive and feels that the topic is of high interest.

MAJOR POINTS:

1. The main disappointment with the study, as it currently stands, is the lack of functional analysis using the *Drosophila* embryo. The study begins by documenting the localisation of the different Polo-GFP mutants - Thr182A, Thr182D, NLS7A - driven in the embryo using the Mat-a4-GAL4 driver, in the presence of endogenous Polo. Then, at the very end of the study we are told that, for Thr182D and NLS7A, this leads to a failure to hatch of ~80% and ~90% of embryos. But there is nowhere an indication of what the consequences on mitosis, karyokinesis, cell cycle timing etc are in embryos. There is nice work in Dmel2 cells (Figure 5), but it does seem amiss to leave out an analysis in syncytial embryos. Similarly, the localisation of Polo-GFP NLS7A in embryos is fascinating - it appears to be excluded, post NEB, from centrosomes, kinetochores/centromeres and the spindle and instead localises to the endoplasmic reticulum (Figure 2E). Yet, this is skipped over in the text - all that is mentioned is "NLS7A abolished the nuclear localisation of Polo in prophase" (line 153-4). The authors must have considered what this might mean. In my view, it is imperative to complement the cell biology of Dmel2 cells presented in Figure 5 with an investigation of the functional consequences of expressing the mutated versions of Polo-GFP in the early embryo, correlating this to the sub-cellular localisations.

We thank the reviewer for prompting us to do this work. We have now analyzed the phenotypes resulting from the expression of the different forms of Polo-GFP in embryos, as requested. Our observations are described in the last section of the Results section and are presented in the new Figs. 6E-G and S9. We have found interesting correlations between embryo hatching rates, Polo activity (MPM2 reactivity), localization and mitotic and meiotic phenotypes. Because we observe dominant effects with Polo-GFP WT, NLS7A and T182D, we did not attempt genetic rescue experiments in this system. We do not intend to extend further our phenotypic characterization in this study as its major contributions lay instead in the new mechanism of spatiotemporal regulation of Polo at mitotic entry that we have uncovered, and how this mechanism is required for the regulation of Cdc25^{String} by Polo.

As requested, we have added a description of the intriguing localization pattern of Polo^{NLS7A}-GFP where Fig. 2E is presented in the text. We have also added a quantification of the nuclear localization of the different forms of Polo-GFP in embryos (new Fig S1C).

2. It is not clear from the text, Figure legend or Methods whether the structural figures in Fig 2F and G are the authors re-rendering of published structural data or the result of structural homology modeling of *Drosophila* Polo. Line 859 (Figure 2 legend) says, "structural rendering was performed using PyMOL 1.4 built-in commands", but that's all we're told. There is no mention in the Methods to help clarify.

We have now clarified how Figures 2F, G and 3G were generated in the Methods section and in the legends.

3. It would be good to see the levels of the transgenes expressed in the various cells and tissues - especially in the attempted rescues of the eyeless-GAL4 RNAi in Figure 6. Although we are told

in the Methods that " Insertions on the third chromosome allowing very similar expression levels were selected for experiments", it is possible that the variable level of rescue seen after expressing some of the variants may, at least in part, be due to expression differences. It would help if this could be ruled out.

We verified that all forms of Polo-GFP were expressed at very similar levels in both larvae where expression is driven by ey-Gal4 (new Fig. S8C), and embryos where expression is driven maternally (new Fig. 6E). This result was expected since all lines were generated by site-directed insertions in the same receiving *attP* strain.

4. There is no information on the quantification of images. E.g. Figure 1 C and D, Figure 2D, Figure 4 C - "GFP fluorescence ratio (nucleus/total cell)" - we are not told in Figure legend or Methods how these were calculated. This is important, especially as I'm not convinced that Binucleine 2 delays Polo^{WT}-GFP entry into the nucleus in prophase, as stated in line 142. The data presented in Figure 1D shows only a single time-point as being significantly different between DMSO and Binucleine, suggesting only a minimal delay, if any. Is there any other, stronger data to support the author's conclusion?

We have now clarified the methods used for fluorescence analysis. We have also done statistical tests that reveal a significant delay of Polo^{WT}-GFP nuclear localization in the presence of Binucleine 2. We have also performed the analogous experiment with Polo^{T182D}-GFP. Binucleine 2 does not delay the nuclear localization of Polo^{T182D}-GFP (relative to the DMSO control), which occurs earlier compared with Polo^{WT}-GFP (new Figs. 1D and S2). Binucleine 2 is a selective inhibitor of Drosophila Aurora B (Smurnyy et al, 2010) and we have firmly established that it blocks the phosphorylation of Polo at Thr182 by Aurora B (Carmena et al, 2012; Kachaner et al, 2014; this work, new Fig. S2).

MINOR POINTS:

1. In line 47/48, the authors say Polo becomes concentrated on centromeres at late prophase. Yet, in line 82, they say Polo appears on centromeres before NEB. They should be consistent.

The two phrases are synonyms. To clarify, we have re-written the first sentence the following way: "*Polo becomes concentrated on centrosomes from early prophase and appears on centromeres or kinetochores from late prophase, before nuclear envelope breakdown (NEB).*"

2. *Drosophila* needs to be italicised throughout

I started writing *Drosophila* not italicised after I received an instruction from the Journal "Fly" to do so in a paper. They specified that while *Drosophila melanogaster* is a species name and therefore should always be italicised, the word *Drosophila* alone can now be considered a simple noun and as such it should not be italicised. However, I know that many people still write

Drosophila italicised and I am happy to make that change if that is what is preferred for this journal. In the revised manuscript, we have made the change throughout.

3. Line 591 - "drosophila" needs to be "Drosophila", as well as italicised.

We have made the correction.

4. The "Microscopy" in the Methods needs to be more detailed - the use of deconvolution 3D software implied multiple z-images per time-point, but I could not find details of this.

We have now clarified the methods used.

We thank the reviewers once again for their great help in improving our paper.

Reviewers' Comments:

Reviewer #1 (Remarks to the Author):

The revised manuscript by Kachaner et al has been significantly improved and the concerns I had raised in my initial review have been adequately addressed or at least attempted to my technical satisfaction. The new data and their clarifying interpretations have built a much stronger manuscript.

Reviewer #2 (Remarks to the Author):

I would like to thank the authors for the additional work they have carried out, to further enhance the manuscript and answer the points I initially raised.

I take on board that a full characterisation of the consequences on syncytial mitoses of expressing the mutated Polo variants lies outside the main focus of the study. The additional experiments, quantification and the rewriting of the text are clear, and I am happy to recommend publication.

James Wakefield